# Ethanolic extract of *Otostegia fruticosa* induces ROS-dependent apoptosis and reduces migration of MDA-MB-231 cells *in vitro*

Abdulrahman Alasmari[1,2]*, Chellasamy Panneerselvam[1,2], Saravanan Muthupandian[3,4], Goyitom Gebremedhn Gebru[5]*

1 Department of Biology, Faculty of Science, University of Tabuk, Tabuk, Saudi Arabia, 2 Biodiversity Genomic Unit, Faculty of Science, University of Tabuk, Tabuk, Saudi Arabia, 3 Department of Medical Laboratory Technology, Faculty of Applied Medical Sciences, University of Tabuk, Tabuk, Saudi Arabia, 4 Prince Fahad bin Sultan Chair for Biomedical Research, University of Tabuk, Tabuk, Saudi Arabia, 5 Department of Medical Microbiology, Tigray Health Research Institute, Mekelle, Tigray, Ethiopia

* ggoyitom@yahoo.com (GGG); ab.alasmari@ut.edu.sa (AA)

## Abstract

Breast cancer (BC) has a highly aggressive and metastatic nature. Specifically, triple-negative breast cancer (TNBC) is highly complicated to treat and recover from. Nowadays, treatment strategies focus on the beneficial nature of the herbal source. *Otostegia fruticosa* gained focus on treating various cancers due to its potential against cancer. This study aimed to investigate the apoptotic and metastasis properties of *Otostegia fruticosa* against triple-negative breast cancer cells (MDA-MB-231). The ethanol extract of *Otostegia fruticosa* exhibited cytotoxic effect on MDA-MB-231 cells, with an $IC_{50}$ value of 23 µg/mL. various tests, including MTT, morphological analysis, fluorescence staining, and Trypan blue exclusion, invasion, migration ability were employed to evaluate the effects of *Otostegia fruticosa* on cell viability and apoptosis. The results showed that *Otostegia fruticosa* induced apoptosis in MDA-MB-231 cells, characterized by the accumulation of reactive oxygen species (ROS), mitochondrial damage, and loss of nuclear potential. Furthermore, *Otostegia fruticosa* treatment reduced cell viability due to excessive ROS generation, as evidenced by DCFH-DA staining. Functional assays revealed that *Otostegia fruticosa* impaired cellular mobility, invasion, and colony-forming ability in MDA-MB-231 cells. The *Otostegia fruticose* significantly suppresses invasion property of breast cancer cells about 50% and significantly reduced migrated cells time dependent manner. Gene expression analysis by qRT-PCR and immunofluorescence showed altered expressions of genes linked to apoptotic (Caspase-3, 8, 9, and Bcl2, Bax, Bid, Cytochrome c, PTEN) and metastasis (MMP-9) protein. Notably, *Otostegia fruticosa* downregulated MMP9 and OPN protein expressions, while upregulating caspase-9 and Cyt-c. Western blot analysis validated these findings, highlighting the involvement of the p-Akt and OPN downregulation. Altogether, this study demonstrates that *Otostegia fruticosa* induces apoptosis and inhibits metastasis in MDA-MB-231

**Data availability statement:** All relevant data are within the manuscript and its Supporting information files.

**Funding:** This article is derived from a research grant funded by the Research, Development, and Innovation Authority (RDIA) - Kingdom of Saudi Arabia, with grant number (13445-Tabuk-2023-UT-R-3-1-SE)..

**Competing interests:** The authors have declared that no competing interests exist.

cells by regulating caspase-3, 8, and 9, Bcl2, Bax, and Bid. Meanwhile, it potentially arrests the translation level of MMP-9 and OPN proteins. In addition, cellular migration and invasion were significantly downregulated, and *Otostegia fruticosa* possesses the anti-tumor effect against the TNBC cells (MDA-MB-231).

## Introduction

Breast cancer is a complex and multifaceted disease characterized by the uncontrolled growth and proliferation of malignant cells in the breast tissue [1,2]. Specifically, triple-negative breast cancer (TNBC) is a highly aggressive and metastatic subtype of breast cancer, characterized by the absence of estrogen receptors, progesterone receptors, and human epidermal growth factor receptor 2 (HER2) overexpression [3–5]. Due to the lack of targeted therapies, TNBC remains a significant challenge in oncology, with limited treatment options and a poor prognosis.

Natural products have garnered attention as potential anticancer agents, owing to their diverse bioactive compounds and minimal side effects [6,7]. *Otostegia fruticosa*, a member of the *Lamiaceae* family, is a traditionally used medicinal plant in various parts of the world, particularly in the Middle East and North Africa [8,9]. This plant has been employed in folk medicine for centuries to treat a range of ailments, including inflammation, larvicidal, fever, and cancer [10,11]. Significant studies have validated its ethno-pharmacological uses, revealing its potential as a source of bioactive compounds with anticancer, antioxidant, antibacterial, and anti-inflammatory properties [9,12–15]. The essential oils and extracts of *Otostegia fruticosa* have been shown to exhibit cytotoxic effects against various cancer cell lines, including breast cancer cells and endothelial cells [13,16]. However, its molecular mechanisms of action on TNBC cells remain poorly understood.

Indeed, TNBC cells also exhibit increased expression of BCL-2 family proteins, which inhibit apoptosis by blocking the release of cytochrome c from mitochondria [17,18]. Additionally, TNBC cells have been shown to activate the PI3K/AKT signaling pathway, which promotes cell survival and inhibits apoptosis [19–21]. The NF-κB signaling pathway is also constitutively activated in TNBC cells, leading to the expression of anti-apoptotic genes and the suppression of pro-apoptotic genes [22]. Furthermore, TNBC cells can develop resistance to apoptosis through the expression of survivin, a member of the inhibitor of apoptosis protein (IAP) family [23].

Furthermore, TNBC exhibits enhanced metastatic potential through the regulation of various signaling pathways, including the epithelial-to-mesenchymal transition (EMT) pathway [24,25]. TNBC cells also exhibit increased expression of matrix metalloproteinases (MMPs), which degrade the extracellular matrix and facilitate cell migration and invasion [26–28]. The PI3K/AKT pathway is also activated in TNBC cells, promoting cell survival, proliferation, and metastasis [29,30]. Additionally, TNBC cells secrete cytokines and chemokines that recruit immune suppressive cells, creating a pre-metastatic niche that facilitates metastasis [31,32]. Osteopontin (OPN) is a multifunctional protein that promotes metastasis in cancer by enhancing cell migration, invasion, and survival [33,34]. Upon binding to αvβ3 integrin, Osteopontin (OPN) triggers the activation of the

PI3K/pAkt/NF-κB signaling cascade, which in turn regulates the NF-κB/ZEB-mediated epithelial-to-mesenchymal transition (EMT) pathway, ultimately promoting tumor progression and metastasis [35]. Elevated OPN expression has been correlated with poor prognosis and increased metastasis in various cancers, including breast, lung, and colon cancer [36–38].

This study aimed to investigate the apoptotic and metastatic effects of *Otostegia fruticosa* on the triple-negative breast cancer cell line MDA-MB-231, elucidating the underlying molecular mechanisms and exploring its potential as a novel therapeutic agent.

## Materials and methods

### Chemicals used

Cell culture reagents, such as Dulbecco's Modified Eagle Medium (DMEM), antibiotics, and fetal bovine serum (FBS), were procured from American suppliers (Invitrogen, Carlsbad, CA, USA). Fluorescent probes, including Acridine orange, Ethidium bromide, Rhodamine-123 (Rho-123), 2',7'- Dichlorofluorescin diacetate (DCFH-DA), Propidium Iodide (PI), and Hoechst-33342, were purchased from Thermo Fisher Scientific (USA). Primary antibodies (Caspase-9 (#9502) (1:1000 µL), BCL2 (#3498) (1:1000 µL), Cyt-c (#11940) (1:1000 µL), MMP-9 (#3852) (1:1000 µL), OPN (#88742) (1:1000 µL), pAkt (#9271) (1:1000 µL), and Actin B (#4967) (1:2000 µL)) and Secondary antibody (#7056 & #7054/1:5000 µL) utilized in western blot analysis were obtained from Cell Signaling Technology (USA).

### Cell culture method

The MDA-MB-231 human breast cancer cell line was obtained from the American Type Culture Collection (ATCC-HTB-26, USA). Cells were cultured in a humidified atmosphere at 37°C with 5% $CO_2$, using Dulbecco's Modified Eagle's Medium (DMEM) supplemented with 10% fetal bovine serum (FBS) and 1% antibiotic mixture (streptomycin and penicillin) (Invitrogen, Carlsbad, CA, USA). In contrast, HEK-293 normal cells, also sourced from ATCC-CRL-1573 (USA), were maintained according to the recommended protocol provided by the supplier.

### Preparation of *Otostegia fruticosa* ethanol extract

The ethanol extract used in this study was prepared from *Otostegia fruticosa* plants collected from Tabuk, Saudi Arabia. The plant's identity was verified by a botanist, Prof. Fahd Al-Zuaiber from the Department of Biology, University of Tabuk, Tabuk, Saudi Arabia, and the voucher specimens were deposited in the Herbarium of the Department of Biology with voucher no UT/OF/2023/Tabuk/KSA. The collected plant leaves were air-dried in the dark for seven days, then ground into a fine powder. A 100-gram sample of this powder was subjected to ethanol solvent (60% of EtOH/250mL) extraction using a Soxhlet apparatus to yield the ethanol extract under 60–80 °C [39]. After that, Soxhlet running (48 hours) was terminated, and the resulting extract solution was lyophilized (BIFD-001 Lyophilizer Freeze, BR Biochem, India). Finally, the product was stored at −80°C (Deep freezer) until required for experimentation.

### Cytotoxicity evaluation using MTT (3-(4, 5-Dimethylthiazol-2-yl)-2, 5-diphenyltetrazolium bromide) assay

The $IC_{50}$ value of the *Otostegia fruticosa* ethanol extract was evaluated using the MTT assay [40]. Briefly, $1 \times 10^4$ MDA-MB-231 and HEK-293 cells were plated in 96-well plates and incubated overnight. The following day, the culture medium was replaced with fresh DMEM containing varying concentrations (0, 5, 10, 15, 20, 23, 25, 30, 35, 40, 45, and 50 µg/mL) of the *Otostegia fruticosa* extract. After 24 hours of incubation, 20 µl of MTT reagent (5 mg/mL in PBS) was added to each well, followed by a 4-hour incubation. The formazan crystals were then dissolved in 200 µl of DMSO. The absorbance was measured at 595 nm using a multi-well plate reader. Cell viability was calculated using the formula:

$$\% \text{ of cell viability } = \text{ OD of the test/ OD of the control}$$

## Assessment of apoptotic cells via Trypan blue exclusion

Apoptotic cell determination was conducted using the Trypan blue exclusion method [41,42]. MDA-MB-231 cells ($1 \times 10^5$ cells/well) were treated with *Otostegia fruticosa* for 24 and 48 hours. Post-treatment, cells were trypsinized, harvested, and resuspended. Apoptotic cells were then counted using a hemocytometer and Trypan blue dye. The percentage of apoptotic cells was calculated and plotted to illustrate the treatment-induced apoptosis.

## Cell viability analysis

The MTT assay was employed to evaluate the effect of *Otostegia fruticosa* on cell viability. A range of concentrations (0, 23, and 46 μg/mL) was tested across multiple time points (0, 12, 24, and 48 hours) to determine the impact of *Otostegia fruticosa* on cell survival. This test highlights the effect of $IC_{50}$ value and higher doses of *Otostegia fruticosa* on TNBC cells at different time points.

## Morphological analysis

Phase-contrast microscopy was used to investigate cellular morphology changes. MDA-MB-231 cells were exposed to the $IC_{50}$ concentration of *Otostegia fruticosa* for 24 hours. Following treatment, cells were examined using an Accu-Scope EXI-310 microscope at 20 × magnification, and images were captured for further analysis.

## AO/EtBr staining for cell viability

Acridine orange (10 μg/mL) and Ethidium Bromide (10 μg/mL) dual staining technique used to detect the live and dead cells in the experimental group [43,44]. Thus, fluorescence microscopy was employed to evaluate the viability of MDA-MB-231 cells treated with *Otostegia fruticosa*. MDA-MB-231 cells ($1 \times 10^5$ cells/well) were cultured in six-well plates and allowed to adhere overnight. *Otostegia fruticosa* was then added to the cells for 24 and 48 hours. After treatment, cells were stained with AO/EtBr (1:1 ratio) dual staining solution in the dark for 10 minutes. Fluorescence microscopy (Accu-Scope EXI-310) was used to visualize the cells at 20 × magnification, and images were captured.

## Intracellular Reactive Oxygen Species (ROS) measurement

The 2',7'-Dichlorofluorescin diacetate (DCFH-DA) assay was used to evaluate intracellular reactive oxygen species (ROS) levels in MDA-MB-231 cells [45,46]. The amount of ROS molecules potentially converts DCFH-DA into DCF molecule as a fluorescent one. This denoted the presence of the ROS molecule in the MDA-MB-231 cells. Thus, cells ($1 \times 10^5$ cells/well) were cultured in six-well plates and allowed to adhere. Cells were then treated with *Otostegia fruticosa* at the $IC_{50}$ concentration for 24 and 48 hours (Prolonged $IC_{50}$ effect). Following treatment, cells were incubated with DCFH-DA solution in the dark for 30 minutes. After washing, cells were visualized using a fluorescence microscope (Accu-Scope EXI-310) equipped with a green filter, and images were captured.

## Mitochondrial membrane potential analysis (ΔΨm)

The mitochondrial membrane potential (ΔΨm) was evaluated using the Rho-123 staining method to assess mitochondrial integrity [47]. MDA-MB-231 cells ($1 \times 10^5$ cells/well) were cultured in six-well plates and treated with *Otostegia fruticosa* or maintained as untreated controls for 24 and 48 hours. Cells were then washed with phosphate-buffered saline (PBS) and stained with Rho-123. After a 15-minute incubation in darkness, excess dye was removed, and cells were visualized using fluorescence microscopy (Accu-Scope EXI-310) with green light illumination. Images were subsequently captured for analysis.

## Nuclear condensation and fragmentation using Hoechst-33342 staining

The nuclear condensation and fragmentation of MDA-MB-231 cells was examined using the Hoechst-33342 staining method to visualize DNA fragmentation [48]. The Hoechst-33342 solution was prepared by using a $1 \times$ PBS solution by adding Hoechst-33342 salt (10 μg/mL) and served as a working concentration. Cells ($1 \times 10^5$ cells/well) were cultured in six-well plates and allowed to adhere overnight. Cells were then treated with *Otostegia fruticosa* or maintained as untreated controls for 24 and 48 hours. After treatment, cells were stained with Hoechst-33342 dye in the dark for 15 minutes. Fluorescence microscopy (Accu-Scope EXI-310) with a blue filter and $20 \times$ magnification was used to visualize the nuclear morphology.

## Cell viability analysis using Propidium Iodide

To evaluate cell viability, Propidium Iodide (PI), a fluorescent dye (10 μg in mL of $1 \times$ PBS) that selectively stains apoptotic cells, was utilized [49,50]. MDA-MB-231 cells ($1 \times 10^5$ cells/well) were cultured in six-well plates. Cells were then exposed to the $IC_{50}$ concentration of *Otostegia fruticosa* for 24 and 48 hours. After treatment, PI dye was added, and cells were incubated in the dark for 15 minutes. Cell viability was subsequently assessed using fluorescence microscopy (Accu-Scope EXI-310) with a red filter.

## Colony formation analysis

The capacity of cells to form colonies is a critical factor in tumor recurrence. To investigate this, a colony formation assay was performed [51] using crystal violet staining. MDA-MB-231 cells were exposed to *Otostegia fruticosa* and then reseeded for 15 days without interruption. Following this incubation period, colonies were stained with 0.1% crystal violet and counted. The resulting data were used to create a graphical representation of colony formation efficiency.

## Cell invasion analysis using the Transwell assay

The invasive properties of MDA-MB-231 cells were evaluated using a Matrigel-coated Transwell migration insert chamber [52,53]. The upper chamber (8 μm pore size) was pre-coated with Matrigel (2 mg/mL) to mimic the extracellular matrix. *Otostegia fruticosa*-treated cells were seeded on top of the chamber using serum-free DMEM, while the lower chamber contained DMEM (10% fetal bovine serum) as a chemoattractant. After 48 hours at 37ºC incubation, cells that invaded through the Matrigel layer were fixed with methanol (20-minute fixation) and stained with diluted crystal violet (0.1%). Invading cells were then counted using light microscopy at $20 \times$ magnification.

## Cell migration analysis using the wound-healing assay

To investigate the migratory behavior of MDA-MB-231 cells, a wound-healing assay was performed [54–56]. Cells were cultured in six-well plates to form a confluent monolayer. A sterile pipette tip (p-10) was then used to create a wound in the monolayer, and serum-free DMEM containing *Otostegia fruticosa* was added to the wells. The healing process was monitored over time using the Accu-Scope EXI-310 microscope, with images captured ($20 \times$ magnification) at regular intervals. The captured images were then used to detect the migratory field using the ImageJ software. The assay was repeated three independent times (n = 3).

## Quantitative Real-Time PCR (qRT-PCR) analysis

To examine the expression of target genes, qRT-PCR was performed [57]. Total RNA was isolated from *Otostegia fruticosa*-treated and untreated MDA-MB-231 cells using the TRIzol method. Then, 2 μg of total RNA was converted into complementary DNA (cDNA) using the TAKARA cDNA synthesis kit (6110A). Gene expression levels were quantified using Sybr Green-based qRT-PCR with gene-specific primers (listed in Table 1). Beta-Actin served as the internal

**Table 1. Primers were used in this study.**

| S. No | Gene | Forward primer | Reverse Primer | $T_m$ | Product Length (bps) |
|---|---|---|---|---|---|
| 1 | Actin-B | F:5'-TGAAGGCTTTTGGTCTCCCTG −3' | R:5'-ACAAAGTCACACTTGGCCTCA −3' | 59.5 | 118 |
| 2 | Caspase-3 | F:5'-TCCTAGCGGATGGGTGCTAT −3' | R:5'-CTCACGGCCTGGGATTTCAA −3' | 60.2 | 127 |
| 3 | Caspase-8 | F:5'-TTCAGACTGAGCTTCCTGCC −3' | R:5'-GACCAACTCAAGGGCTCAGG −3' | 60.0 | 126 |
| 4 | Caspase-9 | F:5'-AGGCCCCATATGATCGAGGA −3' | R:5'-GGCCTGTGTCCTCTAAGCAG −3' | 59.9 | 139 |
| 5 | BCL2 | F:5'--AAAAATACAACATCACAGAGGAAGT −3' | R:5'-GTTTCCCCCTTGGCATGAGA −3' | 58.5 | 129 |
| 6 | Bax | F:5'-GGAGCAGCCCAGAGGC −3' | R:5'-TTCTTGGTGGACGCATCCTG −3' | 59.8 | 159 |
| 7 | BID | F:5'-TGGGAGACGCTGCCTCG −3' | R:5'-GGAACCGTTGTTGACCTCAC −3' | 60.2 | 138 |
| 8 | CYCS | F:5'-TCGTTGTGCCAGCGACTAAA −3' | R:5'-ACCATGGAGATTTGGCCCAG −3' | 60.1 | 138 |
| 9 | PTEN | F:5'- AGGGACGAACTGGTGTAATGA −3' | R:5'- GGGAATAGTTACTCCCTTTTTGTCT −3' | 58.6 | 125 |
| 10 | PI3K | F:5'-CCCGATGCGGTTAGAGCC −3' | R:5'-TGATGGTCGTGGAGGCATTG −3' | 60.3 | 136 |
| 11 | Akt | F:5'-CAGGATGTGGACCAACGTGA −3' | R:5'-AAGGTGCGTTCGATGACAGT −3' | 59.9 | 137 |
| 12 | MMP9 | F:5'- TCTATGGTCCTCGCCCTGAA −3' | R:5'- GCACAGTAGTGGCCGTAGAA −3' | 59.3 | 198 |

reference gene. The relative quantification of gene expression was calculated using the 2-ΔΔCt method by performing three independent experiments (n = 3).

## Immunocytochemistry analysis

The expression of MMP9 protein in MDA-MB-231 cells was assessed using immunocytochemistry [58]. After the treatment, cells were fixed with 4% paraformaldehyde and gently washed with 1% phosphate-buffered saline (PBS). Cells were then incubated overnight with a primary antibody targeting MMP9 (5 µL/ 5000 µL of 1 × PBS). The next day, the primary antibody was removed, and cells were incubated with an Alexa®Fluor 647-conjugated secondary antibody for 1 hour. Following washing, cells were visualized using fluorescence microscopy (Accu-Scope EXI-310).

## Immunoblotting analysis

To investigate protein expression, immunoblotting was performed [59]. Total cellular proteins were extracted from MDA-MB-231 cells treated with or without *Otostegia fruticosa* (IC$_{50}$ value) using RIPA buffer. Protein concentrations were determined using the Lowry method. Equal amounts of protein (70 µg of protein) were then separated by SDS-PAGE (120 minutes) and transferred to a nitrocellulose membrane (180 minutes). The membrane was incubated overnight with a specific primary antibody, followed by a 4-hour incubation with a secondary antibody. After washing, the membrane was developed using BCIP/NBT substrate for 1 minute. Then, the extra chromogenic substrate was removed by gentle washing and documented.

## Statistical analysis

Results are presented as mean ± standard deviation for three independent experiments (n = 3). Statistical significance was evaluated using the following thresholds: ****$p < 0.0001$, ***$p < 0.001$, **$p < 0.01$, *$p < 0.05$, with "ns" indicating non-significance. One-way and two-way analysis of variance (ANOVA) tests were employed to compare differences between groups by performing Turkey's and Sidak's multiple test.

## Results

### *Otostegia fruticosa* exhibits cytotoxic activity against MDA-MB-231 cells

This study examined the cytotoxic effects of *Otostegia fruticosa* on MDA-MB-231 and HEK-293 (general cytotoxic selectivity cell) cells using the MTT assay. MDA-MB-231 triple-negative breast cancer cells were exposed to various

concentrations in a multi-well plate, revealing a concentration-dependent cytotoxic response. The results confirmed that *Otostegia fruticosa* induced significant cell death. The IC$_{50}$ value was determined to be 23 µg/mL after 24 hours. Additionally, immortalized human embryonic kidney (HEK-293) cells were evaluated, showing an IC$_{50}$ value of 45 µg/mL at 24 hours. These findings indicate that *Otostegia fruticosa* exhibits cytotoxic activity against MDA-MB-231 cells, particularly damaging their cytoplasm (Fig 1a and 1b).

### *Otostegia fruticosa* treatment triggers cytotoxicity and reduces cell viability

To assess cytotoxicity and cell viability loss in MDA-MB-231 cells, trypan blue exclusion and time-course MTT assays were employed. The results demonstrated a dose- and time-dependent relationship, where increasing concentrations of *Otostegia fruticosa* significantly enhanced cell death compared to the control group. Notably, a 48-hour incubation period yielded a higher proportion of dead cells compared to 24 hours. Furthermore, dose- and time-dependent analysis of cell viability revealed a marked decline, as illustrated in Fig 1c and 1d.

### MDA-MB-231 cells exhibit morphological changes upon treatment with *Otostegia fruticosa*

Microscopic examination revealed that *Otostegia fruticosa* treatment disrupted the monolayer alignment of MDA-MB-231 cells. The treated cells exhibited pronounced alterations, including membrane blebbing, cytoplasmic bursting, cell detachment, and loss of cellular integrity. In contrast, control cells maintained their morphology and structural integrity. These observations suggest that *Otostegia fruticosa* exerts significant toxicity towards MDA-MB-231 cells, compromising their cellular structure (Fig 1e).

### *Otostegia fruticosa* treatment induced significant cell death in MDA-MB-231 cells

The cytotoxic effects of *Otostegia fruticosa* on MDA-MB-231 cells were evaluated using dual staining with Acridine Orange (AO) and Ethidium Bromide (EtBr). This assay distinguished between live and dead cells, revealing the impact of *Otostegia fruticosa* treatment. Two time-dependent treatment groups (24 and 48 hours) were compared to the control group. Notably, *Otostegia fruticosa*-treated cells exhibited a significant increase in red fluorescence, indicative of dead cells, with approximately 50% of cells displaying red emission at 24 hours. Furthermore, the 48-hour treatment group showed an even higher proportion of dead cells. These findings demonstrate that *Otostegia fruticosa* exerts cytotoxic effects on MDA-MB-231 cells, as evidenced by the increased red fluorescence (Fig 2a).

### MDA-MB-231 cells exhibit elevated ROS levels following *Otostegia fruticosa* exposure

To investigate ROS production, we employed the DCFH-DA assay. The results showed that *Otostegia fruticosa* treatment significantly increased intracellular ROS levels in MDA-MB-231 cells, with notable elevations observed at 24 and 48 hours. In contrast, control cells exhibited minimal green fluorescence, indicating basal ROS levels. These findings suggest that *Otostegia fruticosa* induces intracellular ROS production in triple-negative breast cancer cells, potentially triggering apoptotic signaling pathways. Quantitative analysis confirmed these results, as illustrated in Fig 2b.

### *Otostegia fruticosa* treatment decreases mitochondrial membrane potential in MDA-MB-231 cells

Mitochondria play a pivotal role in cellular energy metabolism. To assess mitochondrial membrane potential, we employed Rhodamine-123 (Rho-123) staining. Notably, *Otostegia fruticosa*-treated MDA-MB-231 cells exhibited a significant loss of mitochondrial membrane potential. Specifically, cells treated for 24 and 48 hours showed a marked decrease in green fluorescence emission compared to control cells (Fig 2c). These findings provide strong evidence that *Otostegia fruticosa* compromises mitochondrial function in MDA-MB-231 cells.

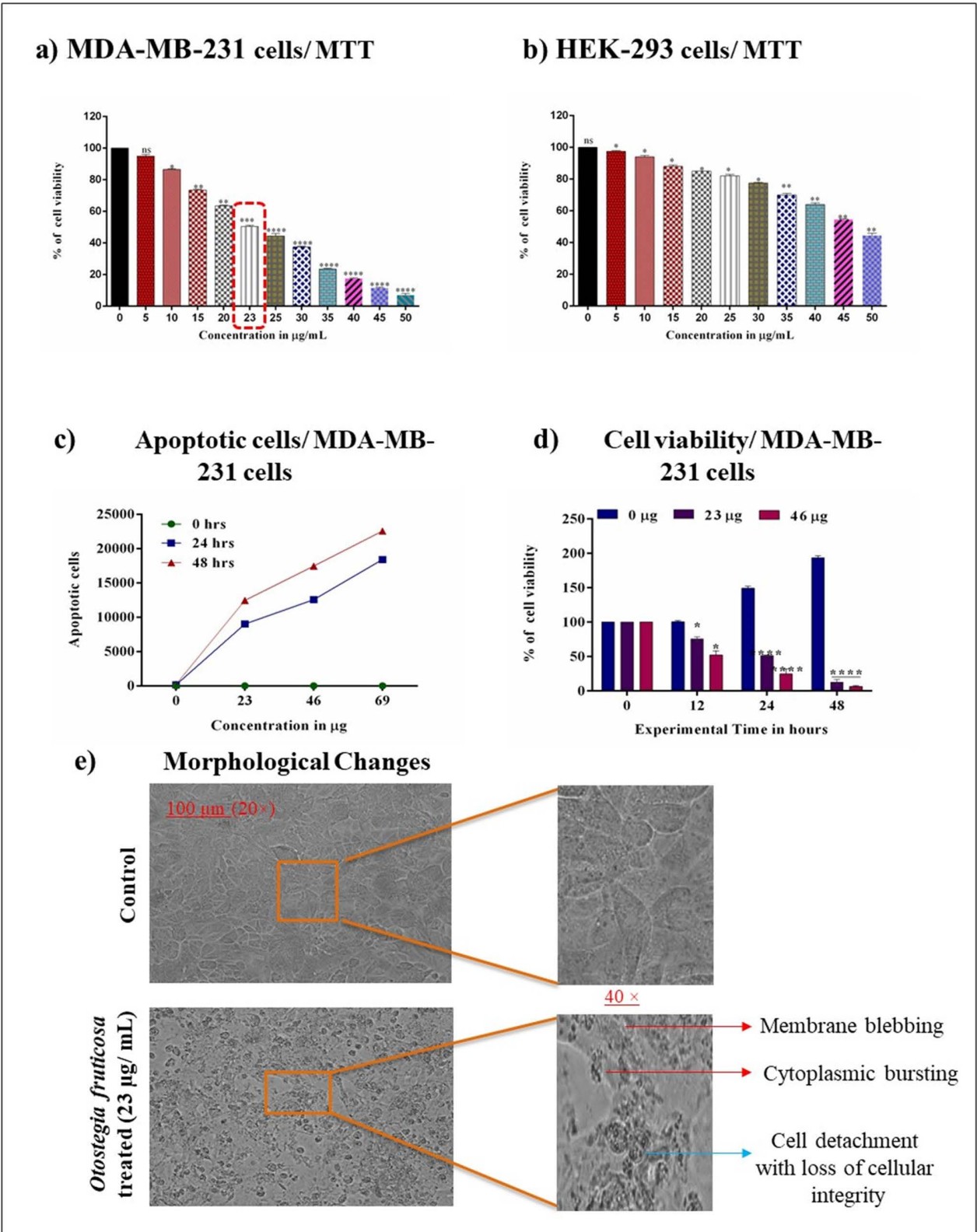

**Fig 1. Illustrates the cytotoxic effects of Otostegia fruticosa on MDA-MB-231 cells.** The results show that Otostegia fruticosa exhibits cytotoxic activity against MDA-MB-231 cells **(a)**, while its cytotoxicity in HEK-293 cells is also assessed **(b)**. Furthermore, Otostegia fruticosa induces apoptosis

in MDA-MB-231 cells in a time- and dose-dependent manner **(c)**, and similarly affects cell viability **(d)**. Morphological changes in MDA-MB-231 cells with and without Otostegia fruticosa treatment are visualized using phase contrast microscopy **(e)**. All data represent mean ± SD for three independent detections (n = 3) with statistical significance determined by one-way ANOVA followed by Dunnett's multiple comparison test, where ****$p < 0.0001$, ***$p < 0.001$, **$p < 0.01$, *$p < 0.05$, and ns indicate non-significance.

### *Otostegia fruticosa* treatment compromises DNA integrity in MDA-MB-231 cells

Apoptotic events are characterized by nuclear collapse and damage. To investigate this, we performed a DNA fragmentation assay. Our results show that *Otostegia fruticosa* treatment induces nuclear membrane potential loss and degradation. Microscopic analysis revealed a significant increase in blue fluorescence emission in 24 and 48-hour-treated cells, indicating nuclear damage, compared to control cells (Fig 2d). These findings confirm that *Otostegia fruticosa* exerts a detrimental effect on the nucleus of MDA-MB-231 cells.

### *Otostegia fruticosa* treatment induces late apoptosis and inhibits colony formation in MDA-MB-231 cells

To assess late apoptosis, we employed PI staining under microscopy. In contrast to untreated cells, which exhibited low red fluorescence, *Otostegia fruticosa*-treated cells showed increased red emission, indicating membrane potential loss and damaged nuclei (Fig 2e). This suggests that *Otostegia fruticosa* induces late apoptosis and viability loss in triple-negative breast cancer cells. Furthermore, dose-dependent administration of *Otostegia fruticosa* (23 and 46 µg) significantly downregulated colony formation in MDA-MB-231 cells compared to control cells (Fig 2f). Notably, higher concentrations of *Otostegia fruticosa* suppressed colony formation more effectively *in vitro*. These findings confirm that *Otostegia fruticosa* actively inhibits single-cell-based colony formation in triple-negative breast cancer cells.

### *Otostegia fruticosa* treatment inhibits the invasive capacity of MDA-MB-231 cells

Cellular invasion is a crucial step in the formation of secondary tumors, leading to severe consequences for the host. To investigate this, we conducted a Matrigel-based invasion assay. The results showed that *Otostegia fruticosa* administration (23 and 46 µg) significantly suppressed the invasion of MDA-MB-231 cells through the Matrigel matrix. In contrast, control cells actively invaded the Matrigel and attached to the lower part of the insert chamber. The observed results and data quantification confirmed that *Otostegia fruticosa* effectively arrested the invasive potential of triple-negative breast cancer cells (Fig 2g).

### *Otostegia fruticosa* treatment inhibits the migratory capacity of MDA-MB-231 cells

To investigate the migratory capacity of MDA-MB-231 cells, we employed the wound healing assay in the presence of *Otostegia fruticosa*. The assay was conducted at various time points, and results showed that *Otostegia fruticosa*-treated cells exhibited significantly impaired wound closure at 12, 24, and 48 hours. In contrast, control cells displayed time-dependent wound closure. These findings suggest that *Otostegia fruticosa* effectively arrests cellular migration in a 2D cell culture model system (Fig 3a and 3b).

### *Otostegia fruticosa* treatment induces the upregulation of caspase-3 and −9 in MDA-MB-231 cells

Apoptosis markers, including caspase-3 and −9, play crucial roles in regulating apoptotic signaling. We investigated the expression levels of these markers in response to *Otostegia fruticosa* treatment. Notably, qRT-PCR analysis revealed significantly higher expression levels of caspase-3 and −9 mRNA in *Otostegia fruticosa*-treated MDA-MB-231 cells. Specifically, time-dependent treatment with *Otostegia fruticosa* resulted in increased caspase-3 mRNA expression, with levels rising significantly over time. Similarly, caspase-9 mRNA levels also demonstrated a time-dependent increase (Fig 3c). These findings indicate that *Otostegia fruticosa* effectively upregulates caspase-3 and −9 mRNA production in triple-negative breast cancer cells.

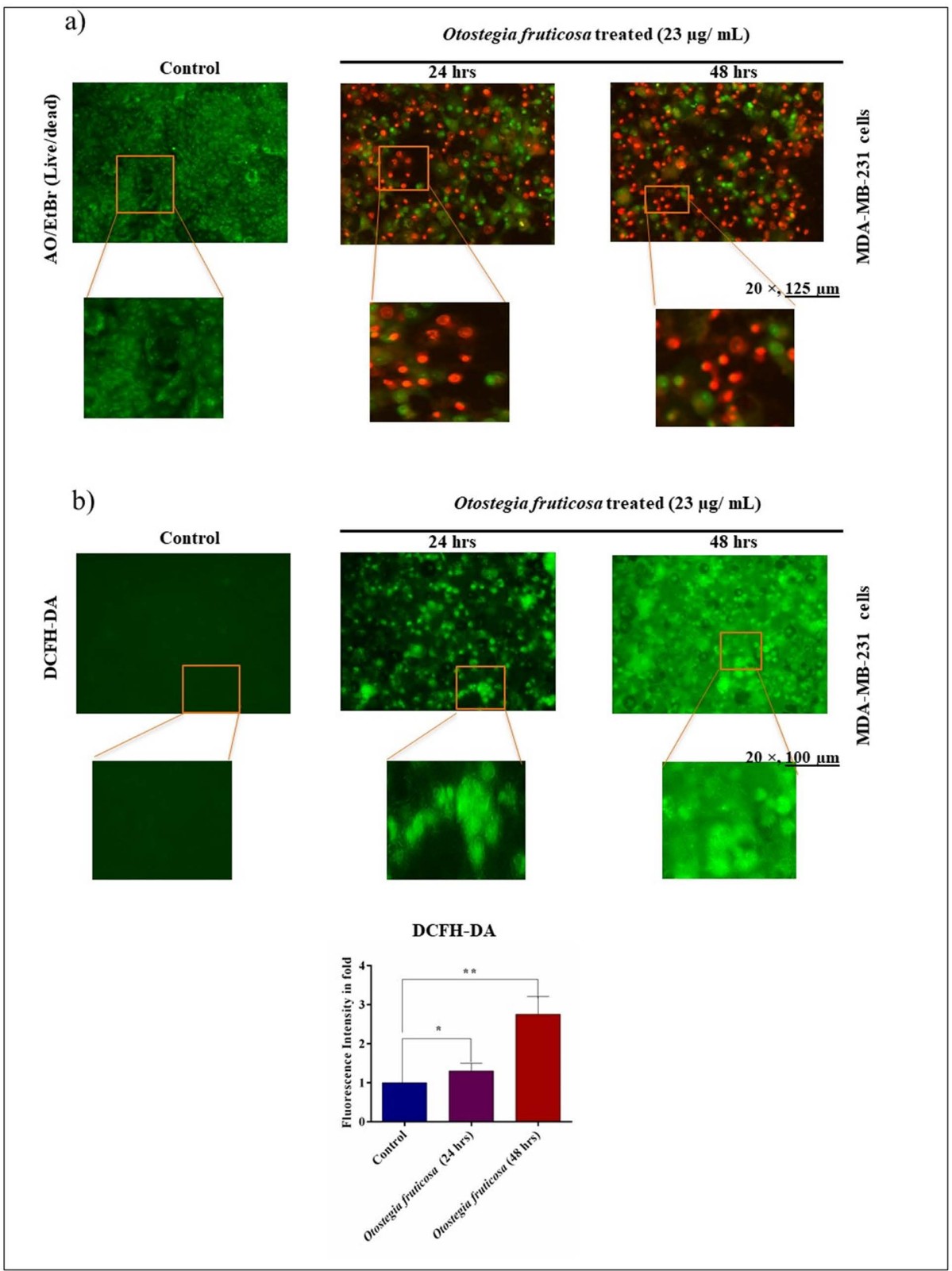

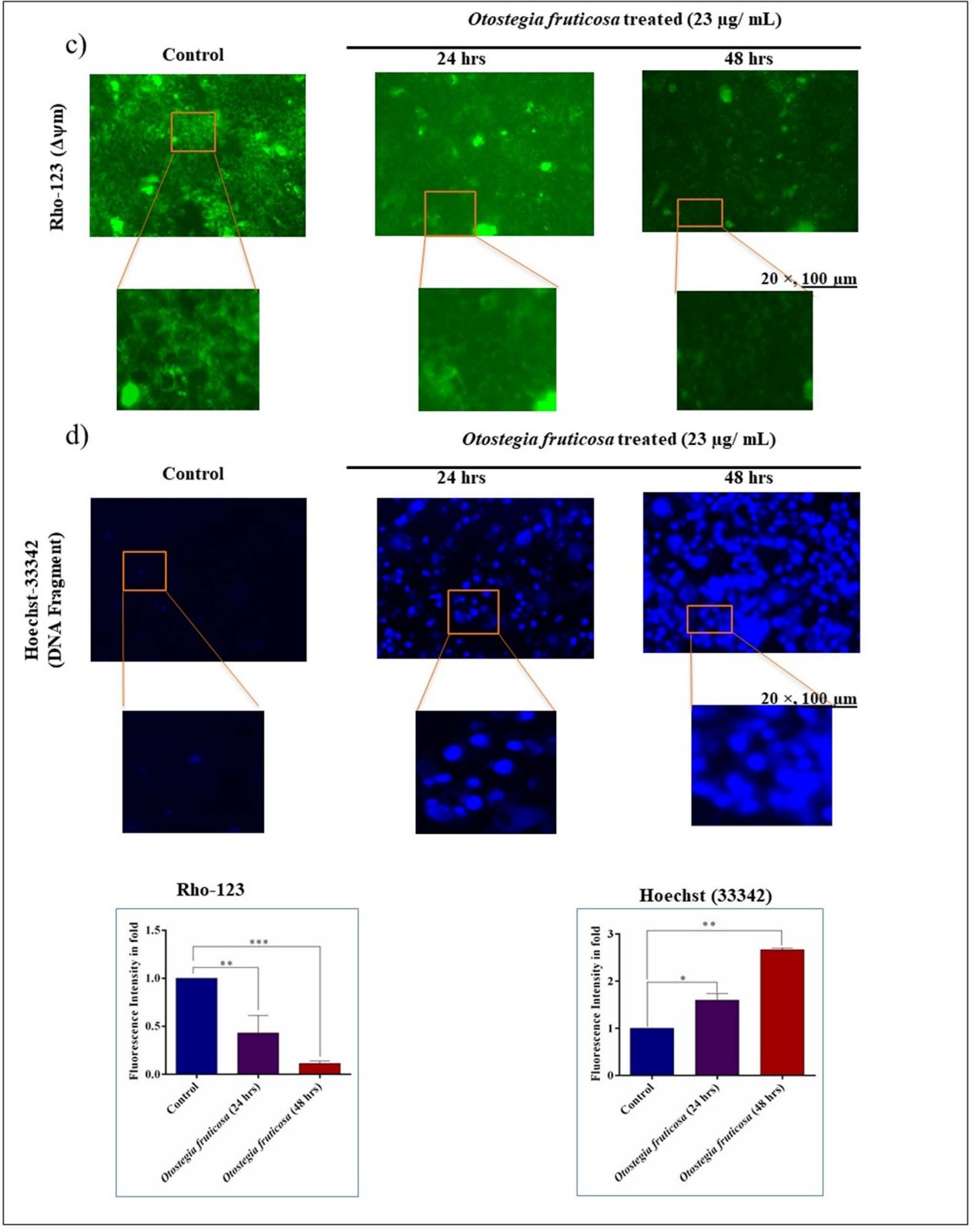

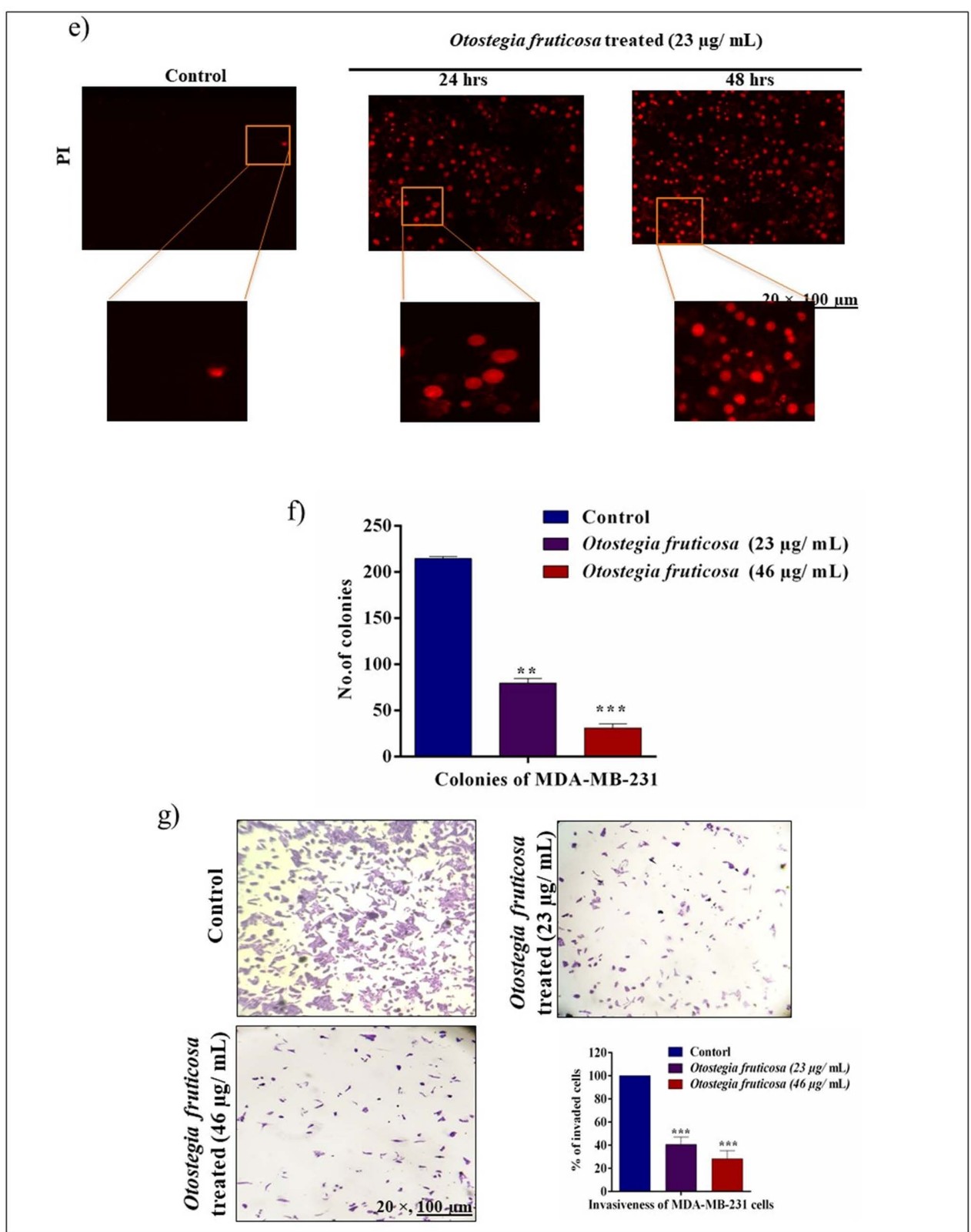

**Fig 2. Otostegia fruticosa induces cytoplasmic changes leading to apoptosis in MDA-MB-231 cells.** The fluorescence microscopy images demonstrate the effects of Otostegia fruticosa treatment on MDA-MB-231 cells. Acridine orange/ethidium bromide (AO/EtBr) staining reveals a shift from green to

red fluorescence in treated cells, indicating apoptosis **(a)**. A significant increase in intracellular reactive oxygen species (ROS) is observed in treated cells, as shown by the enhanced green fluorescence **(b)**. The mitochondrial membrane potential (Δψm) is decreased in treated cells, as evidenced by reduced green fluorescence following Rho-123 staining **(c)**. DNA fragmentation analysis shows increased blue fluorescence in treated cells, indicating nuclear damage **(d)**. Propidium iodide (PI) staining detects a higher number of late apoptotic cells in treated cultures, characterized by red fluorescence **(e)**. Otostegia fruticosa treatment also inhibits colony formation **(f)** and cell invasion **(g)**, as assessed by Transwell invasion assay. Images were captured at 100 µm. Data are presented as mean±SD for three independent detections (n=3) with statistical analysis performed using one-way ANOVA followed by Dunnett's multiple comparison test. Significant differences are denoted by ****$p < 0.0001$, ***$p < 0.001$, **$p < 0.01$, *$p < 0.05$, and ns indicate non-significance.

### *Otostegia fruticosa* treatment modulates the apoptotic pathway and metastasis-related genes in MDA-MB-231 cells

The expression of genes involved in the apoptotic and metastatic networks, including caspase-3/8/9, CYCS, BCL2, BID, Bax, MMP9, PTEN, PI3K, and Akt, was analyzed by qRT-PCR after 24-hour treatment with *Otostegia fruticosa*. Notably, caspase-3/8/9 were significantly upregulated, indicating activation of the apoptotic cascade. Additionally, cytochrome c (CYCS) was highly expressed, suggesting mitochondrial membrane potential loss in MDA-MB-231 cells. The anti-apoptotic gene BCL2 was downregulated, while the pro-apoptotic gene BAX was upregulated. Furthermore, MMP9 expression was significantly downregulated, confirming metastasis suppression (Fig 4). We also investigated the effect of *Otostegia fruticosa* on the PI3K/Akt cell survival signaling pathway. qRT-PCR analysis revealed that PTEN (upregulated), PI3K, and Akt mRNA expressions were significantly suppressed in treated cells compared to control cells (Fig 4). These findings demonstrate that *Otostegia fruticosa* induces apoptosis by inhibiting cell survival and metastasis genes.

### *Otostegia fruticosa* treatment downregulates MMP9 protein expression in MDA-MB-231 cells

MMP9 protein plays a crucial role in tumor invasion, metastasis, and angiogenesis in cancer. To investigate the effect of *Otostegia fruticosa* on MMP9 expression, we employed immunocytochemistry to detect MMP9 in the cytoplasm of MDA-MB-231 cells. The results showed that *Otostegia fruticosa* treatment significantly reduced MMP9 protein levels, as indicated by decreased red fluorescence, whereas the nucleus was stained blue (Fig 5). In contrast, control cells exhibited stronger red fluorescence, suggesting higher MMP9 expression and potential for metastasis.

### *Otostegia fruticosa* induces apoptosis in MDA-MB-231 cells, triggering cell death

The expression of apoptosis-regulated proteins, including Caspase-9, BCL2, and Cyt-c, as well as metastasis-regulated proteins (MMP9 and OPN), and the cell survival signaling protein pAkt, was analyzed in MDA-MB-231 cells treated with *Otostegia fruticosa*. Notably, Caspase-9 expression was significantly upregulated (1.8-fold) compared to untreated cells. Conversely, the anti-apoptotic protein BCL2 was downregulated, while Cyt-c was markedly upregulated. Furthermore, *Otostegia fruticosa* treatment resulted in the downregulation of MMP9 and OPN, indicating inhibition of metastasis in MDA-MB-231 cells. Analysis of the PI3K/Akt cell survival pathway revealed a substantial decrease in phosphorylated Akt (pAkt) levels, suggesting downregulation of this pathway in triple-negative breast cancer cells (Fig 6). Collectively, these findings demonstrate that *Otostegia fruticosa* induces apoptosis and suppresses metastasis in MDA-MB-231 cells by upregulating pro-apoptotic proteins, downregulating anti-apoptotic proteins, and inhibiting the p-Akt protein.

## Discussion

Triple-negative breast cancer treatment typically involves a combination of therapies, including chemotherapy, targeted therapies, and immunotherapies. Chemotherapy remains the backbone of treatment for early and advanced TNBC. Recent approvals include pembrolizumab, an immune checkpoint inhibitor, for patients with high PD-L1 expression, and sacituzumab govitecan, an antibody-drug conjugate [60–63]. Despite these advancements, TNBC treatment faces significant challenges. The lack of reliable biomarkers hinders patient stratification and treatment optimization. Additionally, managing side effects and associated symptoms remains a major concern. Further research is needed to overcome these limitations and develop more effective therapies for TNBC [60,64,65].

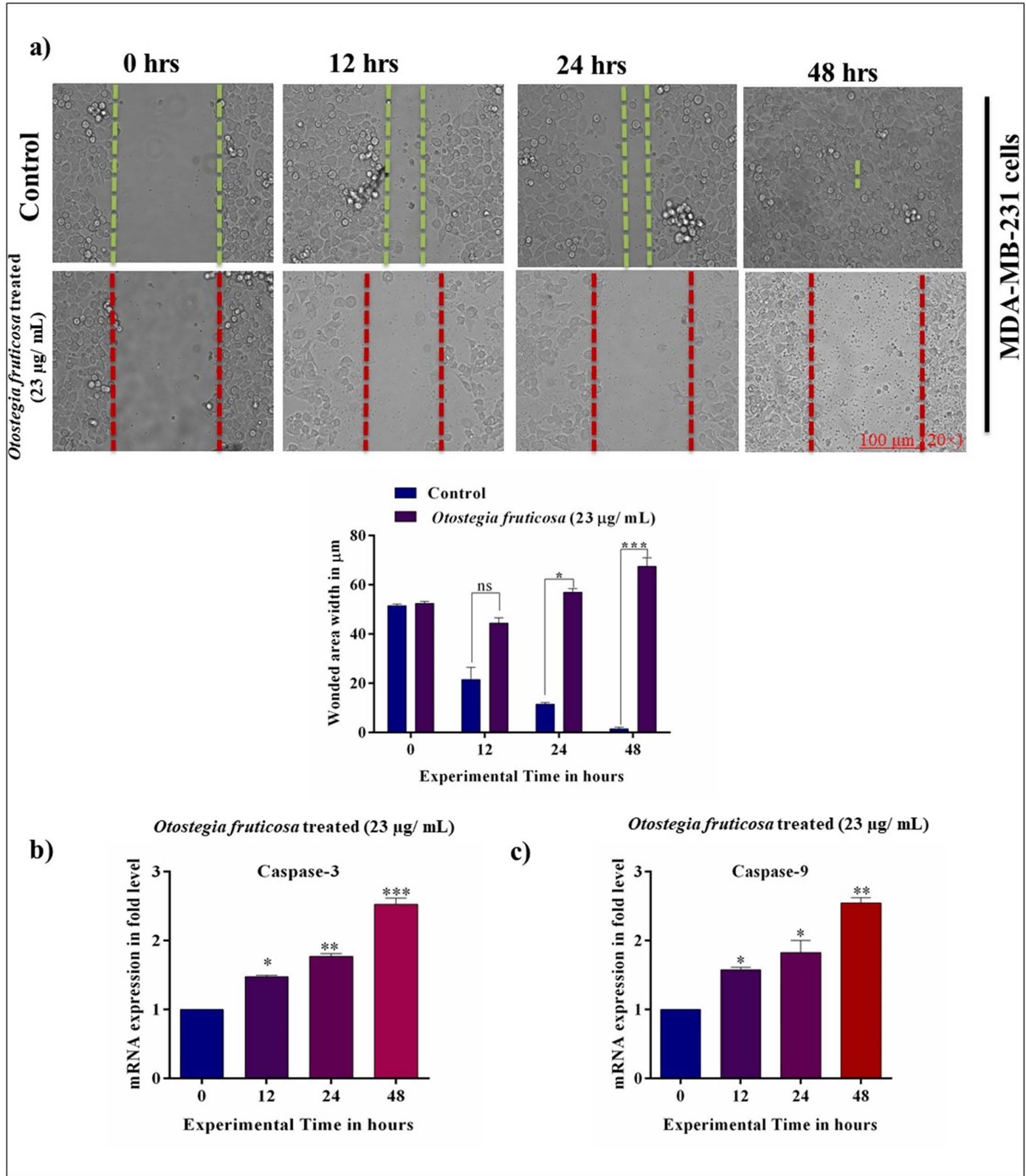

**Fig 3. *Otostegia fruticosa* disrupts cell migration and upregulates caspase mRNA in MDA-MB-231 cells.** The wound healing assay reveals impaired migration efficiency in MDA-MB-231 cells treated with *Otostegia fruticosa*, as demonstrated by time-dependent image capture and

quantification **(a)**. *Otostegia fruticosa* treatment also induces significant upregulation of caspase-3 **(b)** and caspase-9 **(c)** mRNA levels in MDA-MB-231 cells. Images were acquired at 100 μm. Data represent mean±SD for three independent detections (n=3) with statistical analysis performed using one-way ANOVA followed by Dunnett's multiple comparison test. Significant differences are indicated by ****$p<0.0001$, ***$p<0.001$, **$p<0.01$, *$p<0.05$, and ns indicate non-significance.

The present study elucidates the action of *Otostegia fruticosa* on induced apoptosis and metastasis in triple-negative breast cancer cells. Originally, the ethanol extraction of *Otostegia fruticosa* was tested against the triple-negative breast cancer cells MDA-MB-231 cells. It produces cytotoxicity at a concentration of 23 μg/mL in 24 hours [10]. It may be due to the presence of toxic compounds from the extraction [14]. M. N. Ansari et al., 2020 and Hawwal MF et al., 2024 listed that hydro-alcoholic extract of *Otostegia fruticosa* composed of numerous bioactive phytochemicals through GC-Ms analysis such as 2-Methyl-Benzaldehyde, 3,7,11,15-Tetramethyl-2-Hexadecen-1-ol, E AND Z Isomers OF 1-(2,6,6-Trimethyl-1-Cyclohexen-1-YL)-3,4,4-Trimethyl-2-Pent-ene, Santalane, (+)-2-Endo,3-Endo-Dimethylbornane, bornyl formate, myrtenyl formate, β-caryophyllene etc. [8,12]. In particular, β-caryophyllene gained great attention for hepatocellular carcinoma, which arrests HCC cell proliferation actively, with relatively low toxic effects on normal liver cells [66]. The induced cytotoxicity potentially regulates apoptosis by altering the network proteins for mitochondria-mediated cell death [67]. Due to the activated apoptosis, our present results reflected the apoptotic cell production under the *Otostegia fruticosa*. In other words, it is reflected in decreased cell viability in MDA-MB-231 cells. Altogether, it affects the structural integrity of cancer cells.

Medicinal plants possess the active phytochemicals that trigger the intracellular ROS in the cancer cells. Further, it leads to an unstable position in the cytoplasm and creates oxidative stress [68,69]. The oxidative stress negatively regulates the cellular structure including membrane lipids, lipoproteins, and DNA [70–75]. Marking, mitochondrial ROS release during electron transport chain dysfunction causes oxidative damage, mitochondrial permeability transition, and cytochrome c release, culminating in apoptosome formation and apoptosis [76]. These circumstances provoke the impaired ATP production, calcium dysregulation, mPTP opening and activation of apoptosis [77]. Our current study's results align with these published data, demonstrating that *Otostegia fruticosa* induces oxidative stress in the cytoplasm of MDA-MB-231 cells, leading to mitochondrial permeabilization. Cell death was confirmed using AO/EtBr and PI staining. Mitochondrial dysfunction, characterized by elevated ROS production, was detected using DCFH-DA and Rho-123 analysis. Furthermore, Hoechst staining revealed DNA fragmentation, indicating apoptosis.

Furthermore, BCL2 is an anti-apoptotic protein that regulates mitochondrial outer membrane permeabilization (MOMP), preventing the release of cytochrome c and inhibiting caspase activation. Overexpression of BCL2 blocks apoptosis, promoting cell survival, while its downregulation sensitizes cells to apoptosis. Dysregulation of BCL2 is implicated in various cancers, highlighting its potential as a therapeutic target [78–80]. *Otostegia fruticosa* induced BCL-2 downregulation in MDA-MB-231 cells, corroborating established research on the role of BCL-2 in apoptosis regulation. In consideration of pro-apoptotic, BAX plays the vital role in apoptosis that the induced level of BAX promotes the MOMP and releases the cytochrome c, which activates the caspase cascade and sensitizes cells to apoptosis [81], and the same phenomenon was also documented in this study.

The activated level of caspase-9 cleaves and activates executioner caspases (caspase-3 and −7), inducing DNA fragmentation and apoptosis [82,83]. Targeting caspase-9 has shown promise in cancer therapy, particularly in combination with pro-apoptotic agents, such as BH3 mimetics [84,85], and the documented data perfectly authenticates the current findings about caspase-9 upregulation under *Otostegia fruticosa*. Another hallmark of apoptosis is the release of cytochrome c, which is triggered by pro-apoptotic proteins (BAX, BAK) and inhibited by anti-apoptotic proteins (BCL-2, BCL-XL). Upon release, cytochrome c binds to Apaf-1, forming the apoptosome, which activates caspase-9 and initiates the caspase cascade [86,87]. Targeting cytochrome c release has shown promise in cancer therapy, particularly in combination with chemotherapeutics and BH3 mimetics [85,88].

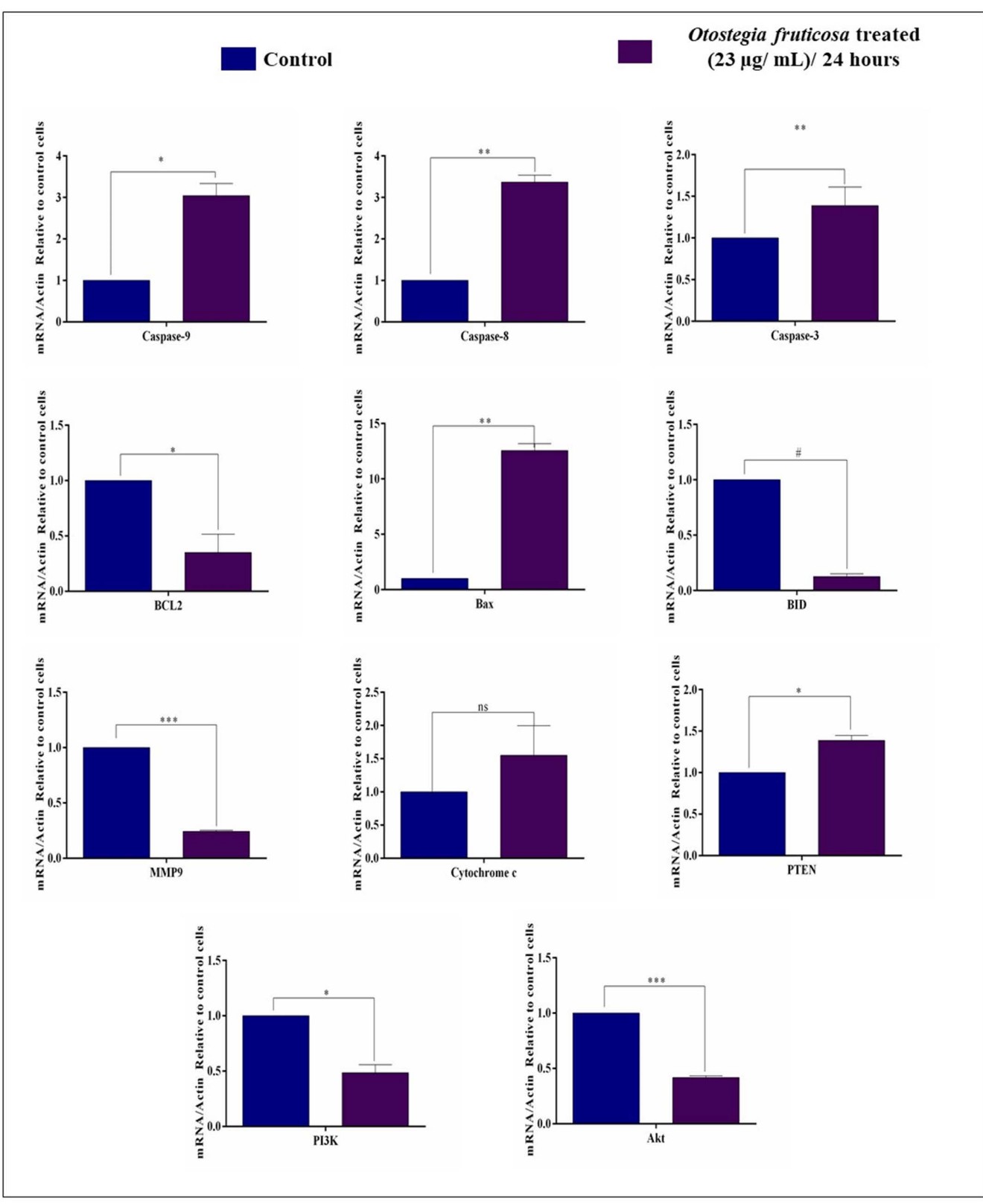

**Fig 4. *Otostegia fruticosa* modulates gene expression to induce apoptosis and suppress metastasis in MDA-MB-231 cells.** Quantitative real-time PCR (qRT-PCR) analysis reveals significant changes in mRNA expression profiles following *Otostegia fruticosa* treatment. Specifically, the data

show upregulation of pro-apoptotic genes (Caspase-3, −8, and −9, Bax, and BID) and downregulation of anti-apoptotic (BCL2) and metastatic (MMP9) genes. Additionally, *Otostegia fruticosa* treatment affects key regulatory genes, including Cytochrome c, PTEN, PI3K, and Akt. Results are presented as mean ± SD for three independent detections (n = 3) with statistical significance determined by one-way ANOVA followed by Dunnett's multiple comparison test (****p < 0.0001, ***p < 0.001, **p < 0.01, *p < 0.05, and ns indicate non-significance).

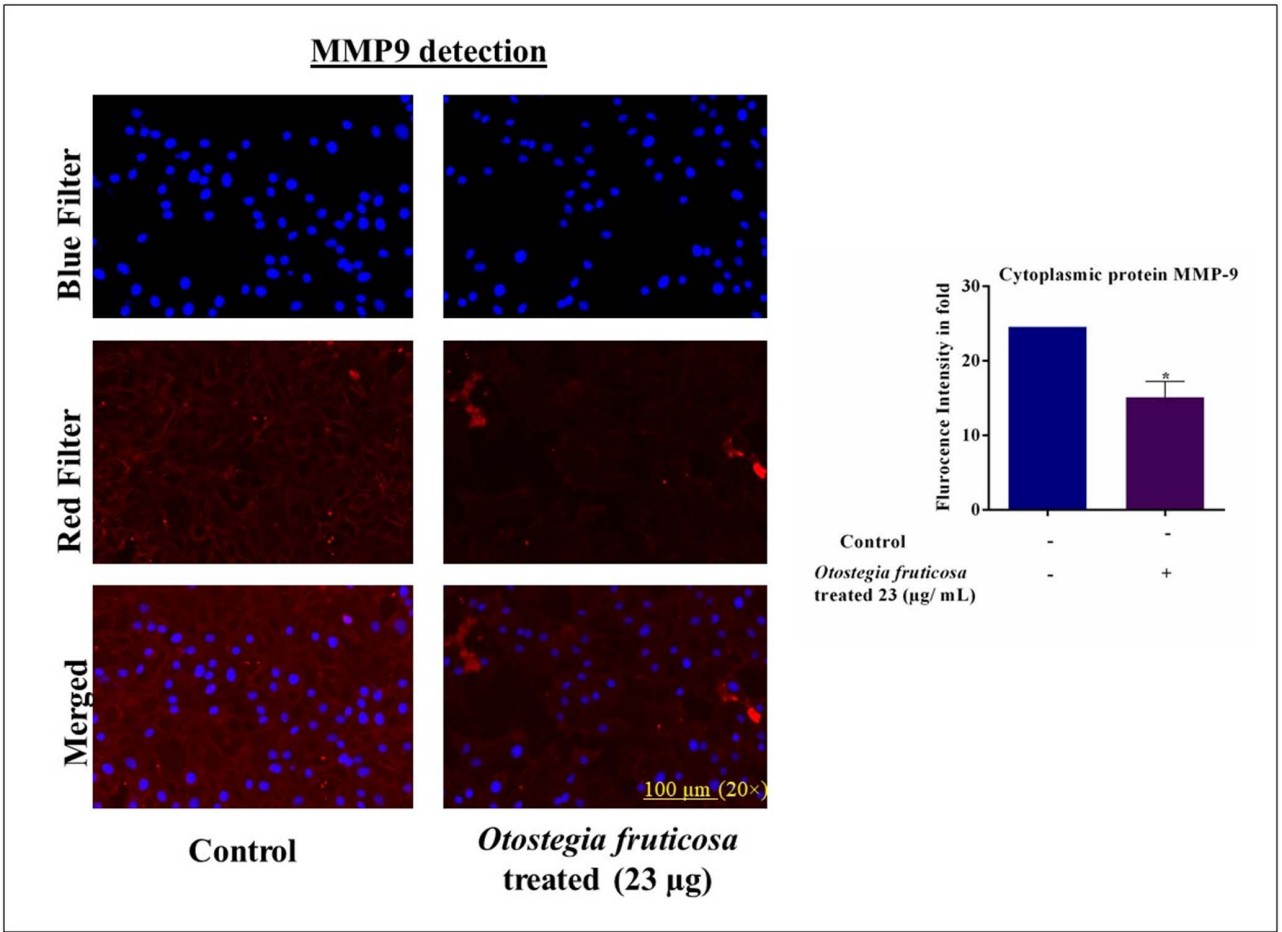

**Fig 5. Otostegia fruticosa downregulates MMP9 protein expression in MDA-MB-231 cells.** Immunocytochemical analysis reveals a decrease in MMP9 protein levels in cells treated with Otostegia fruticosa. Cells were fixed, incubated with MMP9 primary antibody, and detected with Alexa® Fluor-647-conjugated secondary antibody. Fluorescence microscopy images, captured at 100 μm, show reduced red fluorescence in treated cells compared to controls, indicating suppressed MMP9 expression. Quantitative analysis confirms this observation. Results are presented as mean ± SD for three independent detections (n = 3) with statistical significance determined by one-way ANOVA followed by Dunnett's multiple comparison test (****p < 0.0001, ***p < 0.001, **p < 0.01, *p < 0.05, and ns indicate non-significance).

The PI3K/AKT signaling pathway promotes cell survival, proliferation, and metastasis in cancer cells. Activation of PI3K leads to phosphorylation and activation of AKT, inhibiting apoptosis and enhancing cell growth [89]. Hyperactivation of PI3K/AKT is implicated in various cancers, including breast, lung, and colon cancer [90,91]. Targeting PI3K/AKT with inhibitors, such as wortmannin and LY294002, has shown anti-tumor efficacy in preclinical studies [92,93]. Combination therapies involving PI3K/AKT inhibitors and chemotherapeutics or targeted agents hold promise for improved cancer treatment outcomes [94]. This study confirms the downregulation of PI3K. Akt while upregulating PTEN in MDA-MB-231 and suppresses the cell survival under the *Otostegia fruticosa*.

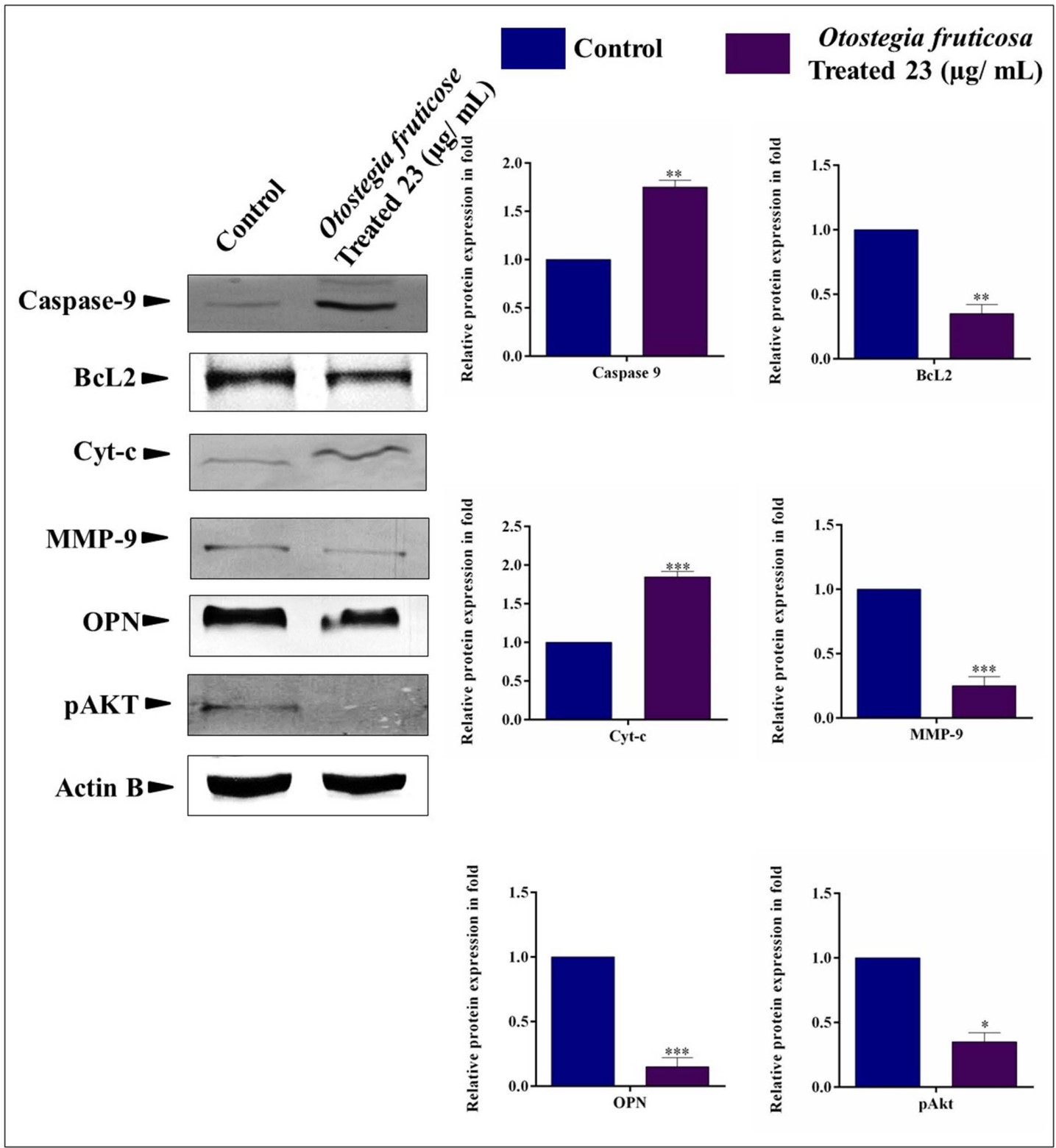

**Fig 6. Otostegia fruticosa modulates protein expression to induce apoptosis and inhibit metastasis in MDA-MB-231 cells.** Western blot analysis reveals changes in protein levels following Otostegia fruticosa treatment. Specifically, the data show increased expression of caspase-9 and cyt-c, and decreased expression of BCL2, MMP-9, Osteopontin (OPN), and phosphorylated AKT (p-AKT). Actin B served as the internal control. Results are

presented as mean±SD for three independent detections (n = 3) with statistical significance determined by one-way ANOVA followed by Dunnett's multiple comparison test (*p < 0.0001, **p < 0.001, **p < 0.01, *p < 0.05, and ns, non-significant).

On the consideration of metastasis, OPN and MMP9 play the crucial role in cancer metastasis. Deeply, Osteopontin/MMP9 plays a vital role in cancer metastasis. OPN enhances cell migration, invasion, and adhesion, while MMP9 degrades extracellular matrix, facilitating tumor cell spread [95–97]. Elevated OPN and MMP9 expression correlates with poor prognosis in various cancers, including breast, lung, and colon cancer [28,33,98]. Targeting OPN and MMP9 with inhibitors, such as OPN-neutralizing antibodies and MMP inhibitors, has shown promise in reducing metastasis and improving cancer treatment outcomes [99,100]. Our tested ethanol extraction, corroborated with stated findings of the OPN/MMP9 axis and authenticated arresting of metastasis with downregulated MMP9/OPN, was noted in wound healing and invasion cellular function in MDA-MB-231 under *Otostegia fruticosa*. The possible mechanism of apoptosis and metastasis arrest is elucidated in Fig 7.

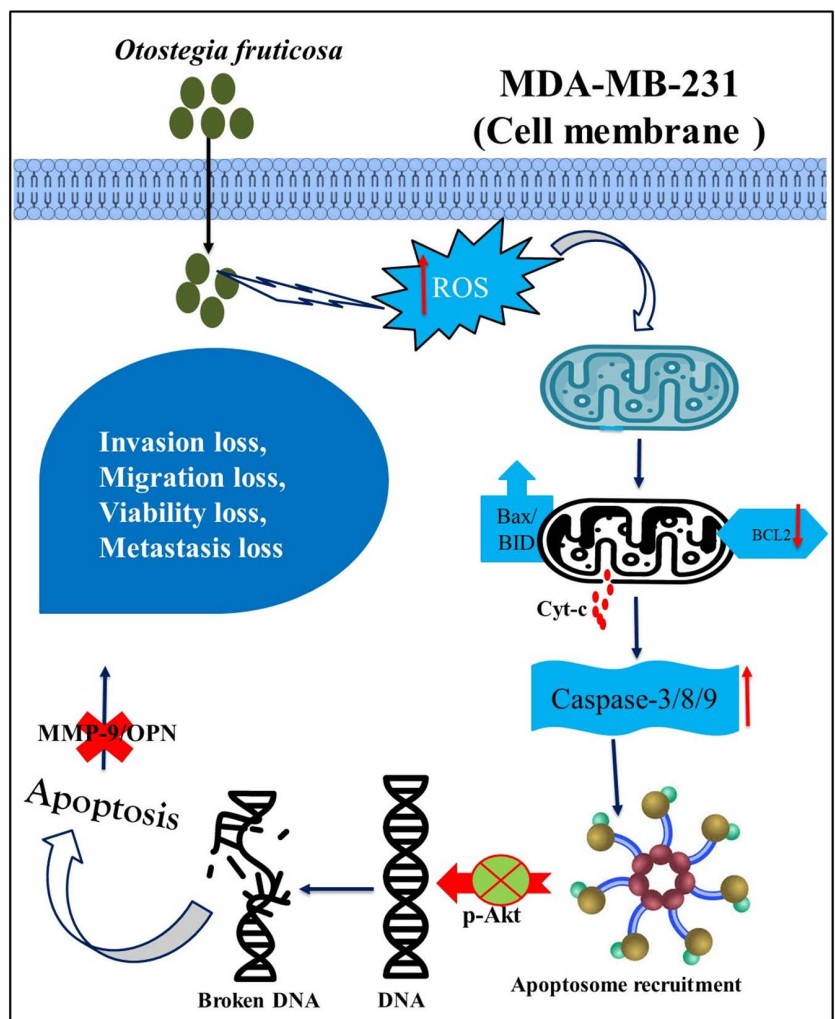

**Fig 7. Schematic illustration of Otostegia fruticosa's mechanism of action.** This diagram depicts the proposed pathway by which Otostegia fruticosa induces apoptosis and inhibits metastasis in Triple Negative breast cancer cells, summarizing the key findings of this study.

## Conclusion

To conclude, *Otostegia fruticosa* extract exhibited significant cytotoxic effects on MDA-MB-231 cells, inducing apoptosis and reducing cell viability. The extract triggered characteristic cytoplasmic changes indicative of apoptosis (damaged cell membrane and blebbing), including increased ROS production, mitochondrial dysfunction, DNA fragmentation (Fragmented nucleus), and apoptotic cell formation. The cellular functional studies demonstrated inhibition of cell invasion and migration. Furthermore, the plant extract modulated the apoptotic gene network, downregulating anti-apoptotic genes (BcL2) and upregulating pro-apoptotic genes (Bax). Protein analysis revealed suppression of metastasis-related proteins, such as MMP-9 and OPN, and activation of apoptosis-related proteins, including caspase-9 and Cyt-c, while inhibiting BCL2 and phosphorylated Akt. These findings collectively confirm the apoptotic effects of *Otostegia fruticosa* extract on MDA-MB-231 cells and confirm the anti-cancer potentiality of *Otostegia fruticosa*. However, future studies should prioritize the identification and isolation of the specific high-functioning phytochemicals responsible for the observed effects, paving the way for advanced preclinical investigations using animal models.

## Author contributions

**Conceptualization:** Abdulrahman Alasmari.

**Data curation:** Abdulrahman Alasmari.

**Formal analysis:** Abdulrahman Alasmari, Chellasamy Panneerselvam, Saravanan Muthupandian.

**Investigation:** Abdulrahman Alasmari, Chellasamy Panneerselvam.

**Methodology:** Abdulrahman Alasmari, Chellasamy Panneerselvam, Saravanan Muthupandian.

**Software:** Abdulrahman Alasmari, Chellasamy Panneerselvam.

**Supervision:** Goyitom Gebremedhn, Saravanan Muthupandian.

**Validation:** Abdulrahman Alasmari, Chellasamy Panneerselvam, Saravanan Muthupandian.

**Visualization:** Abdulrahman Alasmari, Chellasamy Panneerselvam.

**Writing – original draft:** Goyitom Gebremedhn, Abdulrahman Alasmari, Saravanan Muthupandian.

**Writing – review & editing:** Goyitom Gebremedhn, Abdulrahman Alasmari, Saravanan Muthupandian.

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
