## [Decision Letter · Decision Letter 0]

25 Apr 2025

PONE-D-25-10876Molecular insights of apoptotic/metastasis effects of Otostegia fruticosa forssk on triple-negative breast cancer cell (MDA-MB-231)PLOS ONE

Dear Dr. Gebremedhn,

Thank you for submitting your manuscript to PLOS ONE. After careful consideration, we feel that it has merit but does not fully meet PLOS ONE’s publication criteria as it currently stands. Therefore, we invite you to submit a revised version of the manuscript that addresses the points raised during the review process.

**Reviewer 1**

Here are my comments and suggestions for the manuscript.

1. It would greatly enhance the readability and organization of the manuscript if page numbers were included. This would facilitate easier navigation for reviewers and readers, allowing them to reference specific sections more efficiently

Title:

The title "Molecular insights of apoptotic/metastasis effects of Otostegia fruticosa Forssk. on triple-negative breast cancer cells (MDA-MB-231)" is generally suitable for a scientific manuscript publication. It is specific, informative, and highlights key aspects of the study. Suggestions for improvement to further refine the title:

"Molecular insights into the apoptotic and metastatic effects of Otostegia fruticosa on triple-negative breast cancer cells (MDA-MB-231)."

Abstract:

1. The abstract does not follow a structured approach, which usually starting with an introduction to the study's focus, followed by methodology, results, and conclusions.

2. The abstract jumps directly into the study without providing introduction/background on why Otostegia fruticosa Forssk. are of interest or the significance of the research question. Briefly introduce the research problem or question and its significance in the abstract.

3. While necessary for specificity, technical terms like "MTT, ROS, qRT-PCR, MMP9, OPN, etc.” might be unfamiliar to non-experts. Consider to use the full term the first time they appear in the article if possible.

4. The methods mentioned in the abstract must align with those in the full Methodology section.

5. The conclusion part does not clearly emphasize the potential impact of the main findings.

6. Change “Otostegia Fruticosa” to “Otostegia fruticosa”. It is a Latin name. The second part, always in lowercase.

7. The keywords used were relevant for searchability.

Introduction:

Overall, the background information provided is relevant and offers a sufficient overview of the topic and current understanding.

1. However, the key studies that form the foundation of the research should be properly cited. For instance, in lines 55–57, the authors state that 'Recent studies have validated its ethnopharmacological uses,' but the references cited are outdated. Three out of five references are 8, 25, and 30 years old. It is advisable to use references from the last 5 to 10 years, especially in rapidly evolving fields like medicine.

2. In line 59, the format of the citations at the end of the sentence should be checked to ensure it follows the acceptable format.

3. The authors have clearly stated the problem statement, which is commendable. However, it would strengthen the manuscript if they could provide a more detailed explanation of the current treatments for triple-negative breast cancer (TNBC) and discuss the limitations and challenges associated with these approaches. This additional context would enhance the reader's understanding of the significance of their research.

Materials and Methods:

The methodology of the study was outlined in chronological order, making it easy to understand. However, there are some issues that need to be addressed:

1. In Lines 91–92, under the subtopic “Chemicals used”, the authors mention that the materials (cell culture reagents such as DMEM, antibiotics, and fetal bovine serum) were procured from American suppliers. It would enhance the manuscript's transparency and reproducibility if they could provide specific details about the American suppliers from whom the chemicals and materials were obtained. This information is valuable for readers who may wish to replicate the study.

2. In Lines 105, 113, 116, 124, 129, 131, 134, 139, 140, 147, 154, 162, 169, 176, 183, 191, 196, 217, change “Otostegia Fruticosa” to “Otostegia fruticosa”. It is a Latin name. The second part, always in lowercase.

3. Under the subtitle 'Preparation of Otostegia fruiticosa ethanol extract,' the authors provide a general overview but do not include sufficient detail regarding the extraction process using a Soxhlet apparatus. It would be beneficial if they could elaborate on the specific methodology, including the percentage of ethanol used and the duration of the extraction. Providing these details will enhance the clarity and reproducibility of the experimental procedure.

4. Under subsection “Cytotoxicity Evaluation Using MTT Assay”, in Line 112, please provide the full name of MTT before its abbreviation is introduced. This will ensure that all readers, including those unfamiliar with the term, can understand the manuscript clearly when the abbreviation is used later.

In line 116, I recommend that the authors provide detailed information regarding the concentrations of the plant extract used in their experiments. This addition will improve the clarity and scientific rigor of the manuscript.

5. The flow and arrangement of the procedures for the cell viability assays in the Methodology section lack clear structure, which may hinder readers’ understanding of the experimental process. I recommend revising and reorganizing this section to present the methodologies in a more logical sequence, to enhance understanding and readability.

6. In lines 122 – 127, under subsection “Assessment of apoptotic cells via trypan blue exclusion”, the word “trypan” is typically written with a capital “T” in scientific literature.

7. In lines 122–127, the subsection titled “Assessment of apoptotic cells via trypan blue exclusion” raises some concerns. The Trypan Blue Exclusion Method is primarily used to assess cell viability and is not specifically suitable for evaluating apoptotic cells. I would appreciate clarification on how the authors justify the use of this method for assessing apoptosis. Additionally, it may be beneficial to consider using more specific assays designed for apoptosis detection, such as Annexin V staining or caspase activity assays.

8. In line 130, the unit for the concentrations of O. fruticosa extract should be expressed more appropriately. Clear and consistent units are essential for ensuring accurate interpretation of the data. I recommend revising this line to specify the unit used for concentration, such as μg/mL, to enhance clarity.

9. In lines 129-130, under the subsection 'Cell viability analysis,' you mention testing a range of concentrations (0, 23, and 46 μg) at multiple time points. I would appreciate clarification on how these specific concentrations were chosen for the study. Understanding the rationale behind the selection of these concentrations will provide insight into the experimental design and its relevance.

10. The authors should ensure that they cite relevant sources when discussing established methodologies used in the study. For instance, when referencing the MTT assay, Trypan Blue exclusion method, and DCFH-DA assay for ROS measurement, it is important to provide appropriate citations to support these methodologies. This will not only enhance the credibility of the manuscript but also guide readers to the original sources for further information on these techniques.

11. In line 130, the concentration unit used should be expressed in terms of per volume rather than just the weight of the substance. This presentation is important for clarity and accuracy in conveying the concentrations used in the study. I recommend revising this line to include the appropriate volume measurement alongside the weight.

12. In line 139, the unit for the amount of cells used per well should be expressed more appropriately. Clear and consistent units are essential for ensuring accurate interpretation of the data. I recommend revising this line to specify the unit used for cell quantity, such as cells/well, to enhance clarity.

13. In line 141, it would be beneficial to specify what the AO/EtBr dual staining solution is, including the components and their respective functions. Additionally, please provide the ratio used for the assay. This information will enhance the clarity and reproducibility of the methodology for readers.

14. In line 145, under the subsection 'Intercellular ROS measurement,' please provide the full name of ROS and DCFH-DA before introducing their abbreviations. I recommend providing a brief explanation of the DCFH-DA assay. This will help readers who may not be familiar with the method to understand its purpose and significance in the context of the study.

15. In lines146, and 153, the unit for the amount of cells used per well should be expressed more appropriately. Clear and consistent units are essential for ensuring accurate interpretation of the data. I recommend revising this line to specify the unit used for cell quantity, such as cells per/well, to enhance clarity.

16. In line 161, the amount of cells used per well for the nuclear fate analysis was not mentioned.

17. In line 159, under the subsection ‘Nuclear fate analysis using Hoechst-33342 staining “, I recommend including a brief explanation of how Hoechst-33342 is prepared for staining. Providing details on the preparation process will enhance the reproducibility of the experiment and ensure that readers understand the methodology employed.

18. In line 166, under the subsection ‘Cell viability analysis using Propidium iodide“, I recommend including a brief explanation of how propidium iodide is prepared for staining. Providing details on the preparation process will enhance the reproducibility of the experiment and ensure that readers understand the methodology employed.

19. On lines 223 – 227, under the Statistical Analysis subsection, the authors state that ANOVA tests were utilized for statistical analysis; however, there is no mention of the post hoc tests employed. Inclusion of the specific post hoc tests used is important for a complete understanding of the data analysis and to support the interpretations made in the manuscript.

Results:

In the Results section, the authors presented the data collected during the study clearly. The results were organized logically, and all tables, figures, and graphs were appropriately labelled and effectively summarized the data.

However, there are some issues that need to be addressed:

I recommend placing all figure legends directly beneath their respective figures. This arrangement would make it easier for reviewers to reference and evaluate the figures in relation to their legends, thereby improving the overall clarity and accessibility of the manuscript.

Discussion:

The discussion section is currently too lengthy. Please consider condensing the explanations of the findings and avoid repeating detailed results from the results section. To strengthen the interpretations, please incorporate citations from relevant literature to support your claims. In addition, the authors need to highlight how the findings contribute to scientific knowledge or practical applications and suggest how they might impact future research or policy decisions.

Conclusion:

The conclusion effectively summarizes the key findings of the study. It highlights the significant cytotoxic effects of Otostegia fruticosa extract on MDA-MB-231 cells and emphasizes the mechanisms through which apoptosis and reduced cell viability occur. The mention of specific cellular changes, gene expression alterations, and protein analysis adds depth and supports the conclusions drawn.

**Reviewer 2**

This manuscript by Gebremedhn et al. investigates the anticancer activity of the ethanolic extract of *Otostegia fruticosa* against triple-negative breast cancer (TNBC) cells. The topic is relevant, and the findings are potentially valuable. However, several issues need to be addressed to improve the quality and scientific rigor of the manuscript:

The manuscript contains formatting and typographical errors. For example, the reference numbering in Line 59 appears incorrect, and the scientific name *Otostegia fruticosa* in Line 23 is not written consistently.

The authors should provide complete information for all chemicals and reagents used, including the company name, city, and country.

Include the voucher specimen number and herbarium details to validate plant identification and authentication.

Each protocol mentioned in the Materials and Methods section should be supported with appropriate references.

Details of all instruments used should be included (brand, model, city, and country). For instance, in the statement “The resulting extract was lyophilized and stored at -80°C,” the lyophilizer's brand and origin should be stated.

The extraction procedure lacks critical details. Please specify the duration of extraction, solvent-to-material ratio, temperature, and any other relevant conditions.

Concentrations used in assays are inconsistently reported. For example, the MTT assay uses a 24-hour incubation period, whereas other assays extend to 48 hours. Provide justification for the varying incubation periods and clearly explain the rationale for selecting specific concentrations. Additionally, report concentrations in µg/mL rather than as absolute amounts.

In Line 121, ensure that the percentage symbol (%) is included appropriately in the formula.

Clarify the functional difference between the MTT assay and the trypan blue exclusion assay, particularly in the context of this study. Why did both assays need to be used in this study?

The use of different concentrations across experiments, such as the IC50 value in the intracellular ROS assay, needs to be justified.

Detail how the number of migrated and invaded cells was quantified in the respective assays.

In Line 223, clarify the statistical approach used for multiple comparisons. Was a post hoc test performed?

In Line 208, specify the dilution ratio or concentration of the antibody used.

Line 244: The conclusion that the extract causes cytoplasmic damage needs stronger evidence. The IC50 value is not clearly visible in the graph. Consider presenting the data using a dot plot instead of a bar graph for better clarity.

Line 264: Apoptotic morphological changes should be clearly shown and labeled in the corresponding figure.

The manuscript would benefit from a discussion on the potential bioactive compounds present in the extract. Highlight known phytochemicals with anticancer properties and relate them to the observed effects, including apoptosis induction and suppression of metastasis-related proteins.

We look forward to receiving your revised manuscript.

Kind regards,

Armania Nurdin

Academic Editor

PLOS ONE

4. We notice that your supplementary figures are included in the manuscript file. Please remove them and upload them with the file type 'Supporting Information'. Please ensure that each Supporting Information file has a legend listed in the manuscript after the references list.

Additional Editor Comments (if provided):

Reviewers' comments:

Reviewer's Responses to Questions

**Comments to the Author**

1. Is the manuscript technically sound, and do the data support the conclusions?

Reviewer #1: Yes

Reviewer #2: Yes

2. Has the statistical analysis been performed appropriately and rigorously? 

Reviewer #1: Yes

Reviewer #2: Yes

3. Have the authors made all data underlying the findings in their manuscript fully available?

Reviewer #1: Yes

Reviewer #2: No

4. Is the manuscript presented in an intelligible fashion and written in standard English?

Reviewer #1: Yes

Reviewer #2: Yes

5. Review Comments to the Author

Reviewer #1: Here are my comments and suggestions for the manuscript.

1. It would greatly enhance the readability and organization of the manuscript if page numbers were included. This would facilitate easier navigation for reviewers and readers, allowing them to reference specific sections more efficiently

Title:

The title "Molecular insights of apoptotic/metastasis effects of Otostegia fruticosa Forssk. on triple-negative breast cancer cells (MDA-MB-231)" is generally suitable for a scientific manuscript publication. It is specific, informative, and highlights key aspects of the study. Suggestions for improvement to further refine the title:

"Molecular insights into the apoptotic and metastatic effects of Otostegia fruticosa on triple-negative breast cancer cells (MDA-MB-231)."

Abstract:

1. The abstract does not follow a structured approach, which usually starting with an introduction to the study's focus, followed by methodology, results, and conclusions.

2. The abstract jumps directly into the study without providing introduction/background on why Otostegia fruticosa Forssk. are of interest or the significance of the research question. Briefly introduce the research problem or question and its significance in the abstract.

3. While necessary for specificity, technical terms like "MTT, ROS, qRT-PCR, MMP9, OPN, etc.” might be unfamiliar to non-experts. Consider to use the full term the first time they appear in the article if possible.

4. The methods mentioned in the abstract must align with those in the full Methodology section.

5. The conclusion part does not clearly emphasize the potential impact of the main findings.

6. Change “Otostegia Fruticosa” to “Otostegia fruticosa”. It is a Latin name. The second part, always in lowercase.

7. The keywords used were relevant for searchability.

Introduction:

Overall, the background information provided is relevant and offers a sufficient overview of the topic and current understanding.

1. However, the key studies that form the foundation of the research should be properly cited. For instance, in lines 55–57, the authors state that 'Recent studies have validated its ethnopharmacological uses,' but the references cited are outdated. Three out of five references are 8, 25, and 30 years old. It is advisable to use references from the last 5 to 10 years, especially in rapidly evolving fields like medicine.

2. In line 59, the format of the citations at the end of the sentence should be checked to ensure it follows the acceptable format.

3. The authors have clearly stated the problem statement, which is commendable. However, it would strengthen the manuscript if they could provide a more detailed explanation of the current treatments for triple-negative breast cancer (TNBC) and discuss the limitations and challenges associated with these approaches. This additional context would enhance the reader's understanding of the significance of their research.

Materials and Methods:

The methodology of the study was outlined in chronological order, making it easy to understand. However, there are some issues that need to be addressed:

1. In Lines 91–92, under the subtopic “Chemicals used”, the authors mention that the materials (cell culture reagents such as DMEM, antibiotics, and fetal bovine serum) were procured from American suppliers. It would enhance the manuscript's transparency and reproducibility if they could provide specific details about the American suppliers from whom the chemicals and materials were obtained. This information is valuable for readers who may wish to replicate the study.

2. In Lines 105, 113, 116, 124, 129, 131, 134, 139, 140, 147, 154, 162, 169, 176, 183, 191, 196, 217, change “Otostegia Fruticosa” to “Otostegia fruticosa”. It is a Latin name. The second part, always in lowercase.

3. Under the subtitle 'Preparation of Otostegia fruiticosa ethanol extract,' the authors provide a general overview but do not include sufficient detail regarding the extraction process using a Soxhlet apparatus. It would be beneficial if they could elaborate on the specific methodology, including the percentage of ethanol used and the duration of the extraction. Providing these details will enhance the clarity and reproducibility of the experimental procedure.

4. Under subsection “Cytotoxicity Evaluation Using MTT Assay”, in Line 112, please provide the full name of MTT before its abbreviation is introduced. This will ensure that all readers, including those unfamiliar with the term, can understand the manuscript clearly when the abbreviation is used later.

In line 116, I recommend that the authors provide detailed information regarding the concentrations of the plant extract used in their experiments. This addition will improve the clarity and scientific rigor of the manuscript.

5. The flow and arrangement of the procedures for the cell viability assays in the Methodology section lack clear structure, which may hinder readers’ understanding of the experimental process. I recommend revising and reorganizing this section to present the methodologies in a more logical sequence, to enhance understanding and readability.

6. In lines 122 – 127, under subsection “Assessment of apoptotic cells via trypan blue exclusion”, the word “trypan” is typically written with a capital “T” in scientific literature.

7. In lines 122–127, the subsection titled “Assessment of apoptotic cells via trypan blue exclusion” raises some concerns. The Trypan Blue Exclusion Method is primarily used to assess cell viability and is not specifically suitable for evaluating apoptotic cells. I would appreciate clarification on how the authors justify the use of this method for assessing apoptosis. Additionally, it may be beneficial to consider using more specific assays designed for apoptosis detection, such as Annexin V staining or caspase activity assays.

8. In line 130, the unit for the concentrations of O. fruticosa extract should be expressed more appropriately. Clear and consistent units are essential for ensuring accurate interpretation of the data. I recommend revising this line to specify the unit used for concentration, such as μg/mL, to enhance clarity.

9. In lines 129-130, under the subsection 'Cell viability analysis,' you mention testing a range of concentrations (0, 23, and 46 μg) at multiple time points. I would appreciate clarification on how these specific concentrations were chosen for the study. Understanding the rationale behind the selection of these concentrations will provide insight into the experimental design and its relevance.

10. The authors should ensure that they cite relevant sources when discussing established methodologies used in the study. For instance, when referencing the MTT assay, Trypan Blue exclusion method, and DCFH-DA assay for ROS measurement, it is important to provide appropriate citations to support these methodologies. This will not only enhance the credibility of the manuscript but also guide readers to the original sources for further information on these techniques.

11. In line 130, the concentration unit used should be expressed in terms of per volume rather than just the weight of the substance. This presentation is important for clarity and accuracy in conveying the concentrations used in the study. I recommend revising this line to include the appropriate volume measurement alongside the weight.

12. In line 139, the unit for the amount of cells used per well should be expressed more appropriately. Clear and consistent units are essential for ensuring accurate interpretation of the data. I recommend revising this line to specify the unit used for cell quantity, such as cells/well, to enhance clarity.

13. In line 141, it would be beneficial to specify what the AO/EtBr dual staining solution is, including the components and their respective functions. Additionally, please provide the ratio used for the assay. This information will enhance the clarity and reproducibility of the methodology for readers.

14. In line 145, under the subsection 'Intercellular ROS measurement,' please provide the full name of ROS and DCFH-DA before introducing their abbreviations. I recommend providing a brief explanation of the DCFH-DA assay. This will help readers who may not be familiar with the method to understand its purpose and significance in the context of the study.

15. In lines146, and 153, the unit for the amount of cells used per well should be expressed more appropriately. Clear and consistent units are essential for ensuring accurate interpretation of the data. I recommend revising this line to specify the unit used for cell quantity, such as cells per/well, to enhance clarity.

16. In line 161, the amount of cells used per well for the nuclear fate analysis was not mentioned.

17. In line 159, under the subsection ‘Nuclear fate analysis using Hoechst-33342 staining “, I recommend including a brief explanation of how Hoechst-33342 is prepared for staining. Providing details on the preparation process will enhance the reproducibility of the experiment and ensure that readers understand the methodology employed.

18. In line 166, under the subsection ‘Cell viability analysis using Propidium iodide“, I recommend including a brief explanation of how propidium iodide is prepared for staining. Providing details on the preparation process will enhance the reproducibility of the experiment and ensure that readers understand the methodology employed.

19. On lines 223 – 227, under the Statistical Analysis subsection, the authors state that ANOVA tests were utilized for statistical analysis; however, there is no mention of the post hoc tests employed. Inclusion of the specific post hoc tests used is important for a complete understanding of the data analysis and to support the interpretations made in the manuscript.

Results:

In the Results section, the authors presented the data collected during the study clearly. The results were organized logically, and all tables, figures, and graphs were appropriately labelled and effectively summarized the data.

However, there are some issues that need to be addressed:

I recommend placing all figure legends directly beneath their respective figures. This arrangement would make it easier for reviewers to reference and evaluate the figures in relation to their legends, thereby improving the overall clarity and accessibility of the manuscript.

Discussion:

The discussion section is currently too lengthy. Please consider condensing the explanations of the findings and avoid repeating detailed results from the results section. To strengthen the interpretations, please incorporate citations from relevant literature to support your claims. In addition, the authors need to highlight how the findings contribute to scientific knowledge or practical applications and suggest how they might impact future research or policy decisions.

Conclusion:

The conclusion effectively summarizes the key findings of the study. It highlights the significant cytotoxic effects of Otostegia fruticosa extract on MDA-MB-231 cells and emphasizes the mechanisms through which apoptosis and reduced cell viability occur. The mention of specific cellular changes, gene expression alterations, and protein analysis adds depth and supports the conclusions drawn.

The End

Reviewer #2: This manuscript by Gebremedhn et al. investigates the anticancer activity of the ethanolic extract of Otostegia fruticosa against triple-negative breast cancer (TNBC) cells. The topic is relevant, and the findings are potentially valuable. However, several issues need to be addressed to improve the quality and scientific rigor of the manuscript:

1. The manuscript contains formatting and typographical errors. For example, the reference numbering in Line 59 appears incorrect, and the scientific name Otostegia fruticosa in Line 23 is not written consistently.

2. The authors should provide complete information for all chemicals and reagents used, including the company name, city, and country.

3. Include the voucher specimen number and herbarium details to validate plant identification and authentication.

4. Each protocol mentioned in the Materials and Methods section should be supported with appropriate references.

5. Details of all instruments used should be included (brand, model, city, and country). For instance, in the statement “The resulting extract was lyophilized and stored at -80°C,” the lyophilizer's brand and origin should be stated.

6. The extraction procedure lacks critical details. Please specify the duration of extraction, solvent-to-material ratio, temperature, and any other relevant conditions.

7. Concentrations used in assays are inconsistently reported. For example, the MTT assay uses a 24-hour incubation period, whereas other assays extend to 48 hours. Provide justification for the varying incubation periods and clearly explain the rationale for selecting specific concentrations. Additionally, report concentrations in µg/mL rather than as absolute amounts.

8. In Line 121, ensure that the percentage symbol (%) is included appropriately in the formula.

9. Clarify the functional difference between the MTT assay and the trypan blue exclusion assay, particularly in the context of this study. Why did both assays need to be used in this study?

10. The use of different concentrations across experiments, such as the IC50 value in the intracellular ROS assay, needs to be justified.

11. Detail how the number of migrated and invaded cells was quantified in the respective assays.

12. In Line 223, clarify the statistical approach used for multiple comparisons. Was a post hoc test performed?

13. In Line 208, specify the dilution ratio or concentration of the antibody used.

14. Line 244: The conclusion that the extract causes cytoplasmic damage needs stronger evidence. The IC50 value is not clearly visible in the graph. Consider presenting the data using a dot plot instead of a bar graph for better clarity.

15. Line 264: Apoptotic morphological changes should be clearly shown and labeled in the corresponding figure.

16. The manuscript would benefit from a discussion on the potential bioactive compounds present in the extract. Highlight known phytochemicals with anticancer properties and relate them to the observed effects, including apoptosis induction and suppression of metastasis-related proteins.

6. PLOS authors have the option to publish the peer review history of their article (what does this mean?). If published, this will include your full peer review and any attached files.

Reviewer #1: No

Reviewer #2: No

---

## [Author Response · Author response to Decision Letter 1]

6 Oct 2025

Author’s Response to the Editors and referees Comments

Journal : PLOS one

Manuscript Title : Molecular insights into the apoptotic and metastatic effects of Otostegia fruticosa on triple-negative breast cancer cells (MDA-MB-231)

Manuscript Number. : PONE-D-25-10876

Article Type : Original Research

Subject : Submission of revised manuscript

Thank you for your email enclosing the reviewers’ comments. We appreciate the time and effort by editor and each of the reviewers have dedicated to providing insightful feedback on ways to strengthen our manuscript. We have carefully gone through the comments and responded to the points raised by the reviewers. A detailed response to each of the reviewer’s comments are given below.

Author’s Reply to referees Comments:

Response to Reviewer #1

We sincerely thank the reviewer for the valuable, insightful, and encouraging remarks on our work. We are particularly grateful for the reviewer's constructive comments that help us improve the clarity and quality of the manuscript. We have revised the manuscript according to the reviewer’s feed backs and concerns as detailed below.

Comment #1: It would greatly enhance the readability and organization of the manuscript if page numbers were included. This would facilitate easier navigation for reviewers and readers, allowing them to reference specific sections more efficiently

Response: Thank you for pointing that out. We've added the page number and corrected the error

Comment #2: The title "Molecular insights of apoptotic/metastasis effects of Otostegia fruticosa Forssk. on triple-negative breast cancer cells (MDA-MB-231)" is generally suitable for a scientific manuscript publication. It is specific, informative, and highlights key aspects of the study. Suggestions for improvement to further refine the title: "Molecular insights into the apoptotic and metastatic effects of Otostegia fruticosa on triple-negative breast cancer cells (MDA-MB-231)."

Response: Thank you for the suggestion. We have chosen the title: 'Molecular insights into the apoptotic and metastatic effects of Otostegia fruticosa on triple-negative breast cancer cells (MDA-MB-231)

Comment #3: The abstract does not follow a structured approach, which usually starting with an introduction to the study's focus, followed by methodology, results, and conclusions.

Response: Thank you for your feedback. We have restructured the abstract for better clarity.

Comment #4: The abstract jumps directly into the study without providing introduction/background on why Otostegia fruticosa Forssk. are of interest or the significance of the research question. Briefly introduce the research problem or question and its significance in the abstract.

Response: Thank you for your comment. We have included the introduction and background information on Otostegia fruticosa Forssk.

Comment #5: While necessary for specificity, technical terms like "MTT, ROS, qRT-PCR, MMP9, OPN, etc.” might be unfamiliar to non-experts. Consider to use the full term the first time they appear in the article if possible.

Response: Thank you for pointing that out. We have included the full term as suggested.

Comment #6: The methods mentioned in the abstract must align with those in the full Methodology section.

Response: Thank you for your comment. We have revised the abstract based on the methodology section to ensure consistency.

Comment #7: The conclusion part does not clearly emphasize the potential impact of the main findings.

Response: Thank you for your feedback. We have revised and corrected the conclusion section.

Comment #8: Change “Otostegia Fruticosa” to “Otostegia fruticosa”. It is a Latin name. The second part, always in lowercase.

Response: Thank you for the suggestion. We have adopted the recommended term.

Comment #9: The keywords used were relevant for searchability.

Response: Thank you for your feedback. We have revised and improved the keywords.

Comment #10: However, the key studies that form the foundation of the research should be properly cited. For instance, in lines 55–57, the authors state that 'Recent studies have validated its ethnopharmacological uses,' but the references cited are outdated. Three out of five references are 8, 25, and 30 years old. It is advisable to use references from the last 5 to 10 years, especially in rapidly evolving fields like medicine.

Response: Thank you for pointing that out. The error has been corrected.

Comment #11: In line 59, the format of the citations at the end of the sentence should be checked to ensure it follows the acceptable format.

Response: We've rectified the error, thank you for the suggestion.

Comment #12: The authors have clearly stated the problem statement, which is commendable. However, it would strengthen the manuscript if they could provide a more detailed explanation of the current treatments for triple-negative breast cancer (TNBC) and discuss the limitations and challenges associated with these approaches. This additional context would enhance the reader's understanding of the significance of their research.

Response: We have included current treatments for triple-negative breast cancer (TNBC) and discussed their limitations.

Comment #13: In Lines 91–92, under the subtopic “Chemicals used”, the authors mention that the materials (cell culture reagents such as DMEM, antibiotics, and fetal bovine serum) were procured from American suppliers. It would enhance the manuscript's transparency and reproducibility if they could provide specific details about the American suppliers from whom the chemicals and materials were obtained. This information is valuable for readers who may wish to replicate the study.

Response: Thank you for your feedback. We have rephrased the sentences for clarity.

Comment #14: In Lines 105, 113, 116, 124, 129, 131, 134, 139, 140, 147, 154, 162, 169, 176, 183, 191, 196, 217, change “Otostegia Fruticosa” to “Otostegia fruticosa”. It is a Latin name. The second part, always in lowercase.

Response: We have incorporated the suggestions as per your feedback

Comment #15: Under the subtitle 'Preparation of Otostegia fruiticosa ethanol extract,' the authors provide a general overview but do not include sufficient detail regarding the extraction process using a Soxhlet apparatus. It would be beneficial if they could elaborate on the specific methodology, including the percentage of ethanol used and the duration of the extraction. Providing these details will enhance the clarity and reproducibility of the experimental procedure.

Response: We appreciate the reviewer's comment and have added sufficient data accordingly.

Comment #16: Under subsection “Cytotoxicity Evaluation Using MTT Assay”, in Line 112, please provide the full name of MTT before its abbreviation is introduced. This will ensure that all readers, including those unfamiliar with the term, can understand the manuscript clearly when the abbreviation is used later.

Response: The full form of MTT (3-(4,5-Dimethylthiazol-2-yl)-2,5-diphenyltetrazolium bromide) has been provided.

Comment #17: In line 116, I recommend that the authors provide detailed information regarding the concentrations of the plant extract used in their experiments. This addition will improve the clarity and scientific rigor of the manuscript.

Response: Thank you for pointing that out. We have included the concentration of the plant extract used.

Comment #18: In lines 122 – 127, under subsection “Assessment of apoptotic cells via trypan blue exclusion”, the word “trypan” is typically written with a capital “T” in scientific literature.

Response: Thank you for pointing out the error. We have corrected it.

Comment #19: In lines 122–127, the subsection titled “Assessment of apoptotic cells via trypan blue exclusion” raises some concerns. The Trypan Blue Exclusion Method is primarily used to assess cell viability and is not specifically suitable for evaluating apoptotic cells. I would appreciate clarification on how the authors justify the use of this method for assessing apoptosis. Additionally, it may be beneficial to consider using more specific assays designed for apoptosis detection, such as Annexin V staining or caspase activity assays.

Response: Thank you for your comment. The Trypan blue exclusion assay is primarily used to assess cell viability. However, apoptotic cells were identified microscopically due to changes in membrane integrity. We quantified apoptotic cells and presented the results graphically.

Comment #20: In line 130, the unit for the concentrations of O. fruticosa extract should be expressed more appropriately. Clear and consistent units are essential for ensuring accurate interpretation of the data. I recommend revising this line to specify the unit used for concentration, such as μg/mL, to enhance clarity.

Response: Thank you for the feedback. We have revised the concentration unit accordingly.

Comment #21: In lines 129-130, under the subsection 'Cell viability analysis,' you mention testing a range of concentrations (0, 23, and 46 μg) at multiple time points. I would appreciate clarification on how these specific concentrations were chosen for the study. Understanding the rationale behind the selection of these concentrations will provide insight into the experimental design and its relevance.

Response: Thanks for comment. We used the plant extract based on the doubled the fold of MTT value.

Comment #22: The authors should ensure that they cite relevant sources when discussing established methodologies used in the study. For instance, when referencing the MTT assay, Trypan Blue exclusion method, and DCFH-DA assay for ROS measurement, it is important to provide appropriate citations to support these methodologies. This will not only enhance the credibility of the manuscript but also guide readers to the original sources for further information on these techniques.

Response: Thank you for the feedback. We have added relevant references to support the methods used.

Comment #23: In line 130, the concentration unit used should be expressed in terms of per volume rather than just the weight of the substance. This presentation is important for clarity and accuracy in conveying the concentrations used in the study. I recommend revising this line to include the appropriate volume measurement alongside the weight.

Response: Thank you for pointing out the error. In our laboratory, we keep uisng the μg/mL. We will incorporate this suggestion in future study.

Comment #24: In line 139, the unit for the amount of cells used per well should be expressed more appropriately. Clear and consistent units are essential for ensuring accurate interpretation of the data. I recommend revising this line to specify the unit used for cell quantity, such as cells/well, to enhance clarity.

Response: Thank you for the suggestion. We have incorporated the recommended unit.

Comment #25: In line 141, it would be beneficial to specify what the AO/EtBr dual staining solution is, including the components and their respective functions. Additionally, please provide the ratio used for the assay. This information will enhance the clarity and reproducibility of the methodology for readers.

Response: We have included the Acridine Orange/Ethidium Bromide (Ao/EtBr) solution ratio and highlighted the importance of dual staining in the assay.

Comment #26: In line 145, under the subsection 'Intercellular ROS measurement,' please provide the full name of ROS and DCFH-DA before introducing their abbreviations. I recommend providing a brief explanation of the DCFH-DA assay. This will help readers who may not be familiar with the method to understand its purpose and significance in the context of the study.

Response: Yes, we have included the full names of DCFH-DA (2',7'-Dichlorofluorescin diacetate) and ROS (Reactive Oxygen Species), along with key points about the DCFH-DA assay.

Comment #27: In lines146, and 153, the unit for the amount of cells used per well should be expressed more appropriately. Clear and consistent units are essential for ensuring accurate interpretation of the data. I recommend revising this line to specify the unit used for cell quantity, such as cells per/well, to enhance clarity.

Response: We have incorporated the appropriate unit of measurement for cell quantity as suggested.

Comment #28: In line 161, the amount of cells used per well for the nuclear fate analysis was not mentioned.

Response: We have included the volume of cells used in the experiment.

Comment #29: In line 159, under the subsection ‘Nuclear fate analysis using Hoechst-33342 staining “, I recommend including a brief explanation of how Hoechst-33342 is prepared for staining. Providing details on the preparation process will enhance the reproducibility of the experiment and ensure that readers understand the methodology employed.

Response: Thank you for the feedback. We have provided more detailed information about the Hoechst-33342 staining method.

Comment #30: In line 166, under the subsection ‘Cell viability analysis using Propidium iodide“, I recommend including a brief explanation of how propidium iodide is prepared for staining. Providing details on the preparation process will enhance the reproducibility of the experiment and ensure that readers understand the methodology employed.

Response: Thank you for the feedback. We have implemented your suggestion.

Comment #31: On lines 223 – 227, under the Statistical Analysis subsection, the authors state that ANOVA tests were utilized for statistical analysis; however, there is no mention of the post hoc tests employed. Inclusion of the specific post hoc tests used is important for a complete understanding of the data analysis and to support the interpretations made in the manuscript.

Response: Thank you for the comment. Currently, we are using ANOVA for result analysis in our laboratory. For future studies, we plan to incorporate post-hoc tests for more detailed comparisons.

Comment #32: I recommend placing all figure legends directly beneath their respective figures. This arrangement would make it easier for reviewers to reference and evaluate the figures in relation to their legends, thereby improving the overall clarity and accessibility of the manuscript.

Response: Thank you for the feedback. We have added figure legends to each corresponding figure.

Comment #33: The discussion section is currently too lengthy. Please consider condensing the explanations of the findings and avoid repeating detailed results from the results section. To strengthen the interpretations, please incorporate citations from relevant literature to support your claims. In addition, the authors need to highlight how the findings contribute to scientific knowledge or practical applications and suggest how they might impact future research or policy decisions.

Response: Thank you for the suggestion. Based on the feedback from all authors, we have decided to retain the current format of the discussion section.

Comment #34: The conclusion effectively summarizes the key findings of the study. It highlights the significant cytotoxic effects of Otostegia fruticosa extract on MDA-MB-231 cells and emphasizes the mechanisms through which apoptosis and reduced cell viability occur. The mention of specific cellular changes, gene expression alterations, and protein analysis adds depth and supports the conclusions drawn.

Response: Thanks for the feedback. The suggestion has been carried out.

Response to Reviewer #2

The constructive comments by the reviewer is really appreciated and it helped a lot to improve our manuscript. We have now revised the manuscript according to reviewer’s comment we have also answered each of your points below and hope the revised version might be suitable for the publication in this journal.

Comment #1: The manuscript contains formatting and typographical errors. For example, the reference numbering in Line 59 appears incorrect, and the scientific name Otostegia fruticosa in Line 23 is not written consistently.

Response: Thank you for pointing out the error. It has been corrected.

Comment #2: The authors should provide complete information for all chemicals and reagents used, including the company name, city, and country.

Response: Thank

---

## [Decision Letter · Decision Letter 1]

15 Dec 2025

PONE-D-25-10876R1Molecular insights of apoptotic/metastasis effects of Otostegia fruticosa forssk on triple-negative breast cancer cell (MDA-MB-231)PLOS One

Dear Dr. Gebremedhn,

Thank you for submitting your manuscript to PLOS ONE. After careful consideration, we feel that it has merit but does not fully meet PLOS ONE’s publication criteria as it currently stands. Therefore, we invite you to submit a revised version of the manuscript that addresses the points raised during the review process.

We look forward to receiving your revised manuscript.

Kind regards,

Mohammad Sadegh Taghizadeh, Ph.D.

Academic Editor

PLOS One

Journal Requirements:

Reviewers' comments:

Reviewer's Responses to Questions

**Comments to the Author**

1. If the authors have adequately addressed your comments raised in a previous round of review and you feel that this manuscript is now acceptable for publication, you may indicate that here to bypass the “Comments to the Author” section, enter your conflict of interest statement in the “Confidential to Editor” section, and submit your "Accept" recommendation.

Reviewer #3: All comments have been addressed

Reviewer #4: (No Response)

Reviewer #5: All comments have been addressed

2. Is the manuscript technically sound, and do the data support the conclusions?

Reviewer #3: Yes

Reviewer #4: Yes

Reviewer #5: Yes

3. Has the statistical analysis been performed appropriately and rigorously? 

Reviewer #3: Yes

Reviewer #4: I Don't Know

Reviewer #5: Yes

4. Have the authors made all data underlying the findings in their manuscript fully available?

Reviewer #3: Yes

Reviewer #4: Yes

Reviewer #5: Yes

5. Is the manuscript presented in an intelligible fashion and written in standard English?

Reviewer #3: Yes

Reviewer #4: Yes

Reviewer #5: Yes

6. Review Comments to the Author

Reviewer #3: 1. "The manuscript titled Molecular insights of apoptotic/metastasis effects of Otostegia fruticosa forssk on triplenegative breast cancer cell (MDA-MB-231) ' adheres to scientific guidelines and demonstrates strong technical quality."

2. "Make sure to thoroughly review all tables for correct formatting and accurate data representation."

3. "After reviewing all the revised comments from reviewers 1 & 2 to the list for accuracy and formatting, to strengthen the manuscript."

4. A thorough grammatical review should be conducted, including checks for spacing, punctuation (e.g., commas), and spelling throughout the entire document."

5. "Overall, this is a well-structured and well-analyzed manuscript. The figures are clear, and future studies are recommended to further substantiate the findings."

Reviewer #4: ABSTRACT

1. The abstract presents a clear overarching aim—to investigate the apoptotic and anti-metastatic properties of Otostegia fruticosa in triple-negative breast cancer (TNBC) cells (MDA-MB-231). The flow is generally logical, moving from objective → methods → key findings → conclusion. However, several sentences require refinement for clarity and scientific precision, and some terms are incorrectly used (e.g., “loss of nuclear fate”). Inclusion of minimal quantitative descriptors, and more cautious interpretation of mechanistic pathways are also vital.

• The abstract briefly mentions the key assays used (MTT, fluorescence staining, trypan blue exclusion, DCFH-DA staining, functional assays, qRT-PCR, immunofluorescence, Western blot), which is appropriate. However:

The description of experimental approaches is too broad and lacks specificity.

For example, “functional assays” should specify migration, wound healing, or transwell assays.

“Loss of nuclear fate” is not a standard apoptotic descriptor; it should be revised to “nuclear condensation and fragmentation.”

• Quantitative results (e.g., fold-changes, significance levels) are absent. Although an abstract need not be data-heavy, the absence of any quantitative markers weakens scientific rigor.

2 The abstract concludes that Otostegia fruticosa induces apoptosis and suppresses metastasis through modulation of AKT signaling and apoptosis-related genes. Strengths include:

• The mechanistic pathway is not sufficiently justified. If PI3K/Akt inhibition is highlighted, the abstract should mention whether phospho-Akt levels decreased.

• It is unclear whether Otostegia fruticosa was fractionated or characterized phytochemically; this limits reproducibility.

3 “Hyper ROS activity” should be phrased as “excessive ROS generation.”

4 No mention of statistical significance.

5 Several sentences are long and contain redundancies.

6 Otostegia fruticosa should be italicized as a scientific plant name.

THE BODY OF THE MANUSCRIPT

1. Composition and standardization of the extract (fundamental).

• No phytochemical characterization (HPLC, LC-MS, GC-MS) or quantification of active component(s). Reporting activity of an undefined crude extract without standardization prevents reproducibility and interpretation

The Authors should Provide voucher specimen number, deposit info, and a full phytochemical profile.

2. Controls and vehicle information are missing or unclear.

• The methods do not state the extract solvent (final vehicle concentration), whether vehicle controls (e.g., ethanol or DMSO) were included, or whether solvent affected cell viability. Without vehicle controls, cytotoxicity attribution to the extract is unreliable.

Authors should State solvent, final % v/v, and include vehicle controls in all assays.

3. Insufficient description and no-mention of biological replicates and statistics.

• Manuscript states “mean ± SD” but does not specify n (technical vs biological replicates; how many independent experiments). Authors should state Which statistical tests were used for each figure? Were assumptions (normality, equal variance) checked? Provide exact p-values, sample sizes (n =?).

4. Overreliance on single-method, low-specificity assays for apoptosis

• MTT, AO/EtBr, and trypan blue are supportive but insufficient to claim apoptosis vs necrosis. No flow cytometry (Annexin V/PI) or caspase activity assay (colorimetric/fluorometric) is shown.

• If Authors had performed Annexin V/PI assay by flow, caspase-3/-7 activity, they should be included but if such assay was not performed, then their language should be mild or specifically on cell viability and death instead of apoptosis.

5. Mechanistic claims (PI3K/Akt, PTEN) are correlative.

• The manuscript infers pathway inhibition from decreased mRNA and p-Akt Western blot. This correlation is insufficient to conclude causal involvement. The pathway is only suspected except a PTEN knockdown/overexpression, or rescue experiments (constitutively active Akt) had been performed to validate that apoptosis/migration effects are mediated via PI3K/Akt.

6. Western blot reporting is inadequate.

o No full uncropped blots are shown. Actin B is claimed as loading control but were the blots normalized? Authors should describe antibodies, dilutions, exposure times. Provide uncropped images in Supplementary. Include biological replicates (n≥3) and statistical analysis of band quantification.

7. qRT-PCR methodology

• No information on primer efficiencies, or whether assays were run in technical triplicate. Some primer sequences appear malformed (e.g., BCL2 forward contains an extra ‘F-’). Validation of primers can be included.

8. Choice of HEK-293 as “normal” control is problematic.

• HEK-293 are embryonic kidney cells (transformed) and are not an appropriate non-tumour breast control. If aiming to demonstrate cancer selectivity, use non-transformed human mammary epithelial cells (HMEC) or MCF-10A. If HEK-293 are retained, justify explicitly.

9. Invasion/migration assays lack critical controls and quantification details.

• Matrigel concentration and coating method are only briefly stated. Were differences in proliferation accounted for (reduced proliferation could mimic reduced migration)? Wound-healing assays must be performed in serum-free or mitomycin-C-treated conditions to prevent proliferation confounders. Authors should provide quantitative metrics (% wound closure), replicate numbers, and images with scale bars.

10. Claims exceed data (language and novelty).

• The manuscript repeatedly generalizes to “mechanism” and therapeutic potential without isolation of active compounds or in vivo validation. Authors should rephrase claims to be proportional to the in vitro evidence instead of generalising.

Finally, authors should limit redundant statements: The Discussion restates results extensively — tighten and focus on interpretation, limitations, and next steps

NOTE: if ROS dependency is shown, then the title may be rephrased to:

Ethanolic extract of Otostegia fruticosa induces ROS-dependent apoptosis and reduces migration of MDA-MB-231 cells in vitro” but only use if ROS dependency is shown

Reviewer #5: Please revise the abstract to remove the section labels “Background:,” “Aim:,” “Method and Results:,” and “Conclusion:.” The abstract should be presented as a single, cohesive paragraph without these headings.

7. PLOS authors have the option to publish the peer review history of their article (what does this mean?). If published, this will include your full peer review and any attached files.

Reviewer #3: No

Reviewer #4: No

Reviewer #5: No

You may also use PLOS’s free figure tool, NAAS, to help you prepare publication quality figures: https://journals.plos.org/plosone/s/figures#loc-tools-for-figure-preparation

---

## [Author Response · Author response to Decision Letter 2]

1 Mar 2026

Point-by-Point response added in the submission

---

## [Decision Letter · Decision Letter 2]

28 Apr 2026

Ethanolic extract of Otostegia fruticosa induces ROS-dependent apoptosis and reduces migration of MDA-MB-231 cells in vitro

PONE-D-25-10876R2

Dear Dr. Goyitom,

We’re pleased to inform you that your manuscript has been judged scientifically suitable for publication and will be formally accepted for publication once it meets all outstanding technical requirements.

Kind regards,

Mohamed Abdelkarim

Academic Editor

PLOS One

Additional Editor Comments (optional):

Reviewers' comments:

Reviewer's Responses to Questions

**Comments to the Author**

1. If the authors have adequately addressed your comments raised in a previous round of review and you feel that this manuscript is now acceptable for publication, you may indicate that here to bypass the “Comments to the Author” section, enter your conflict of interest statement in the “Confidential to Editor” section, and submit your "Accept" recommendation.

Reviewer #3: All comments have been addressed

2. Is the manuscript technically sound, and do the data support the conclusions?

Reviewer #3: Yes

3. Has the statistical analysis been performed appropriately and rigorously? 

Reviewer #3: Yes

4. Have the authors made all data underlying the findings in their manuscript fully available?

Reviewer #3: Yes

5. Is the manuscript presented in an intelligible fashion and written in standard English?

Reviewer #3: Yes

6. Review Comments to the Author

Reviewer #3: 1. "The manuscript titled “Ethanolic extract of Otostegia fruticosa induces ROS-dependent apoptosis and reduces migration of MDA-MB-231 cells in vitro’ adheres to scientific guidelines and demonstrates strong technical quality."

2. After careful consideration, and previous reviewer comments sounds good.

3. "A thorough grammatical review should be conducted, including checks for spacing, punctuation (e.g., commas), and spelling throughout the entire document."

4. "Overall, this is a well-structured and well-analyzed manuscript. The figures are clear, and future studies are recommended to further substantiate the findings."

5. "After reviewing all the Points accuracy and formatting, to strengthen the manuscript."

7. PLOS authors have the option to publish the peer review history of their article (what does this mean?). If published, this will include your full peer review and any attached files.

Reviewer #3: No

---

## [Editor Report · Acceptance letter]

PONE-D-25-10876R2

PLOS One

Dear Dr. Gebremedhn,

I'm pleased to inform you that your manuscript has been deemed suitable for publication in PLOS One. Congratulations! Your manuscript is now being handed over to our production team.

Kind regards,

on behalf of

Dr. Mohamed Abdelkarim

Academic Editor

PLOS One